# High-altitude hypoxia exposure inhibits erythrophagocytosis by inducing macrophage ferroptosis in the spleen

Wan-ping Yang[1†], Mei-qi Li[1†], Jie Ding[1], Jia-yan Li[1], Gang Wu[2,3], Bao Liu[2,3], Yu-qi Gao[2,3], Guo-hua Wang[1,4]*, Qian-qian Luo[1]*

[1]Department of Physiology and Hypoxic Biomedicine, Institute of Special Environmental Medicine and Co-innovation Center of Neuroregeneration, Nantong University, Nantong, China; [2]College of High-Altitude Military Medicine, Institute of Medicine and Hygienic Equipment for High Altitude Region, Army Medical University, Chongqing, China; [3]Key Laboratory of Extreme Environmental Medicine and High-Altitude Medicine, Ministry of Education of China, Chongqing, China; [4]Department of Neurosurgery, Southwest Hospital, Army Medical University, Chongqing, China

*For correspondence:
wgh036@hotmail.com (GW);
qianqianluo@ntu.edu.cn (Q-qianL);
wgh036@hotmail.com (G-hW)

†These authors contributed equally to this work

**Abstract** High-altitude polycythemia (HAPC) affects individuals living at high altitudes, characterized by increased red blood cells (RBCs) production in response to hypoxic conditions. The exact mechanisms behind HAPC are not fully understood. We utilized a mouse model exposed to hypobaric hypoxia (HH), replicating the environmental conditions experienced at 6000 m above sea level, coupled with in vitro analysis of primary splenic macrophages under 1% $O_2$ to investigate these mechanisms. Our findings indicate that HH significantly boosts erythropoiesis, leading to erythrocytosis and splenic changes, including initial contraction to splenomegaly over 14 days. A notable decrease in red pulp macrophages (RPMs) in the spleen, essential for RBCs processing, was observed, correlating with increased iron release and signs of ferroptosis. Prolonged exposure to hypoxia further exacerbated these effects, mirrored in human peripheral blood mononuclear cells. Single-cell sequencing showed a marked reduction in macrophage populations, affecting the spleen's ability to clear RBCs and contributing to splenomegaly. Our findings suggest splenic ferroptosis contributes to decreased RPMs, affecting erythrophagocytosis and potentially fostering continuous RBCs production in HAPC. These insights could guide the development of targeted therapies for HAPC, emphasizing the importance of splenic macrophages in disease pathology.

## eLife assessment

This **useful** study reports that a week or more of hypoxia exposure in mice increases erythropoiesis and decreases the number of iron-recycling macrophages in the spleen, compromising their capacity for red blood cell phagocytosis – reflected by increased mature erythrocyte retention in the spleen. Compared to an earlier version, the study has been strengthened with mouse experiments under hypobaric hypoxia and complemented by extensive ex vivo analyses. Unfortunately, while some of the evidence is **solid**, the work as it currently stands only **incompletely** supports the authors' hypotheses. While the study would benefit from additional experiments that more directly buttress the central claims, it should be of interest to the fields of hemopoiesis and bone marrow biology and possibly also blood cancer.

## Introduction

A plateau is a special environment characterized by low atmospheric pressure and low partial oxygen pressure. Long-term exposure to plateau environments may lead to chronic mountain disease (*Pérez-Padilla, 2022*). High-altitude polycythemia (HAPC) is a common and widespread chronic mountain sickness characterized by excessive erythrocytosis (*Liu et al., 2022*). Hypobaric hypoxia (HH) is the main cause of erythrocytosis, which in turn alleviates hypoxic conditions in tissues under high-altitude (HA) exposure (*Tymko et al., 2020*). In a healthy organism, a balance is maintained between erythropoiesis in the bone marrow (BM) and erythrophagocytosis in the spleen (*Dzierzak and Philipsen, 2013*). Even though HA/HH exposure can disrupt the equilibrium between RBCs formation and clearance, healthy individuals can swiftly establish a new steady state of RBCs homeostasis in chronic hypoxia (*Robach et al., 2018*). This adaptation is a critical response to the altered oxygen availability characteristic of HA environments. However, in patients with HAPC, RBCs homeostasis is disrupted, failing to reach a state of equilibrium, which ultimately leads to a persistent increase in RBCs count (*Dzierzak and Philipsen, 2013*). This deviation from the normative adaptation process implies a pathological deviation from the usual compensatory mechanisms employed under chronic hypoxia conditions (*Yang et al., 2023*). It is well-established that exposure to HA/HH can induce erythropoiesis, yet the pathogenesis of erythrophagocytosis under these conditions remains poorly understood (*Slusarczyk and Mleczko-Sanecka, 2021*).

The spleen plays a crucial role in maintaining erythropoietic homeostasis by effectively clearing impaired and senescent RBCs from circulation (*Qiang et al., 2023*). Particularly, red pulp macrophages (RPMs) within the spleen, serving as primary phagocytes, are responsible for clearing senescent, damaged, and abnormal erythrocytes from circulation to recycle iron (*Wirth et al., 2020*). RPMs initiate the process of RBCs endocytosis and lysosomal digestion into heme, following the recognition of the RBCs earmarked for removal via their cell surface signals (*Slusarczyk et al., 2023*; *Vahedi et al., 2020*). Subsequently, heme oxygenase-1 (HO-1) decomposes the heme into biliverdin, carbon monoxide, and iron (*Nemeth and Ganz, 2021*). Most of the iron recycled from heme is transported out of the cell through a protein called ferroportin (Fpn). This iron then binds to transferrin (Tf), a plasma protein that transports iron in the blood. The iron-transferrin complex interacts with the transferrin receptor (TfR) on the cell membrane, contributing to the formation of RBCs in the erythroid compartment (*Hidalgo et al., 2021*). A portion of the released iron is loaded into cellular ferritin (Ft; including Ft-H and Ft-L). In the presence of oxygen, the Ft-H facilitates the oxidation of $Fe^{2+}$ (ferrous ion) to $Fe^{3+}$ (ferric ion). Subsequently, the Ft-L stores this $Fe^{3+}$ (*Wang et al., 2013*). The Ft-L features a nucleation site, consisting of a cluster of cavity-exposed carboxyl residues, which readily bind to $Fe^{3+}$, thereby simplifying the storage process (*Finazzi and Arosio, 2014*). When the iron metabolism is vigorous, nuclear receptor coactivator 4 (NCOA4) can recognize Ft-H, bring Ft into the lysosomal pathway for degradation, and release iron in Ft for iron utilization in the body (*Bellelli et al., 2016*). However, some iron exists in cells in the form of ferrous iron ($Fe^{2+}$), which is well known to be an important initiator of free radical oxidation (*Yang et al., 2023*). Moreover, the accumulation of large amounts of $Fe^{2+}$ in cells may cause ferroptosis (*von Krusenstiern et al., 2023*).

Despite the extensive study of erythropoiesis under HA/HH conditions, the precise effects of HA/HH on erythrophagocytosis within the spleen remain largely unexplored. Considering the significant role of the spleen in RBC processing and the crucial function of RPMs in iron recycling from RBC clearance, we evaluated the impacts of HA exposure on the spleen/splenic macrophages using an HH-exposed mouse model (simulating 6000 m exposure conditions). In the current study, we sought to further investigate whether HA/HH can influence the erythrocyte disposal and iron recycling processes in macrophages. More specifically, we aimed to determine the potential impact of HA/HH exposure on the integrity of the erythrophagocytosis process. We discovered that exposure to HH triggered ferroptosis in the spleen, particularly in macrophages. This led to a reduction in macrophage numbers, which was subsequently followed by disruptions in erythrophagocytosis and iron recycling within the spleen. These findings may hold clinical significance, particularly in the context of continuous pathological erythrocytosis and the progression of HAPC under HA exposure.

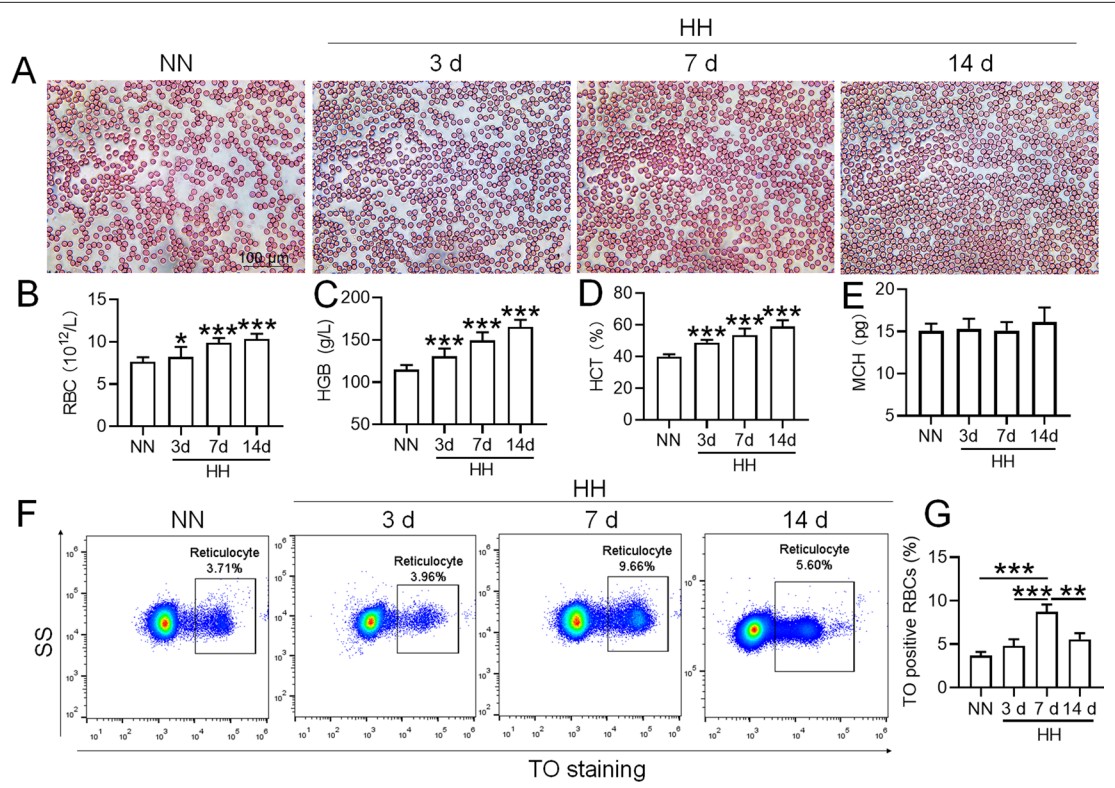

**Figure 1.** HH exposure promotes the induction of erythrocytosis in mice. C57BL/6 mice were subjected to either normobaric normoxia (NN) or hypobaric hypoxia (HH) conditions for durations of 3, 7, and 14 days. After these treatments, blood samples were collected for a comprehensive analysis. (**A**) Morphological evaluation of RBCs was conducted via blood smear examination (Wright staining). Routine hematological assessments were performed, encompassing RBCs counts (**B**), hemoglobin (HGB) levels (**C**), hematocrit (HCT) percentages (**D**), and mean corpuscular hemoglobin (MCH) content (**E**). (**F**) Flow cytometric analysis was employed to identify TO-positive cells, indicative of reticulocytes, in the whole blood samples. (**G**) The proportions of TO-positive RBCs in the blood were depicted in bar graphs. The data are presented as means ± SEM for each group (n=5 per group). Statistical significance is denoted by * p<0.05, ** p<0.01, *** p<0.001, relative to the NN group or as specified.

## Results

### HH exposure promotes erythrocytosis in mice

To mimic 6000 m HA exposure, we placed C58BL/6 mice in an animal hypobaric oxygen chamber and detected the blood indices in the blood of the mice after HH exposure for different times. The blood smear showed that the number of RBCs was increased from 3 to 14 days after HH exposure compared with the NN group (*Figure 1A*). Routine blood tests further confirmed the results in blood smears, which showed that the RBC number (*Figure 1B*), HGB content (*Figure 1C*), and HCT value (*Figure 1D*) were all increased significantly to varying degrees, while MCH was not changed after HH exposure (*Figure 1E*). We further performed flow cytometry using TO staining to detect reticulocytes after 3, 7, and 14 days of HH treatment. The results showed that the proportion of reticulocytes increased after 7 and 14 days of HH exposure (*Figure 1F-G*), and the number of RBCs reached the peak after 7 days of HH exposure. These results suggested that the HH-treated mouse model effectively mimics HA exposure, which promotes erythropoiesis and results in erythrocytosis in mice following HH exposure.

### Spleen inhibits the immoderate increase in RBCs under HH conditions

To determine the roles of the spleen in RBC homeostasis under HA/HH, we investigated the effects of HH on the morphology, volume, and weight of the spleen as well as erythrocyte indices. As shown in *Figure 2*, the spleen volume and weight were decreased significantly after HH exposure for 1 day compared to NN treatment (*Figure 2A-C*). However, the spleen was obviously enlarged from 2 to 14 days after HH exposure (*Figure 2A-C*). The results indicated that the spleen contracted, the stored RBCs in the spleen were released into the blood at 1 day, and the RBCs were produced and/or retained

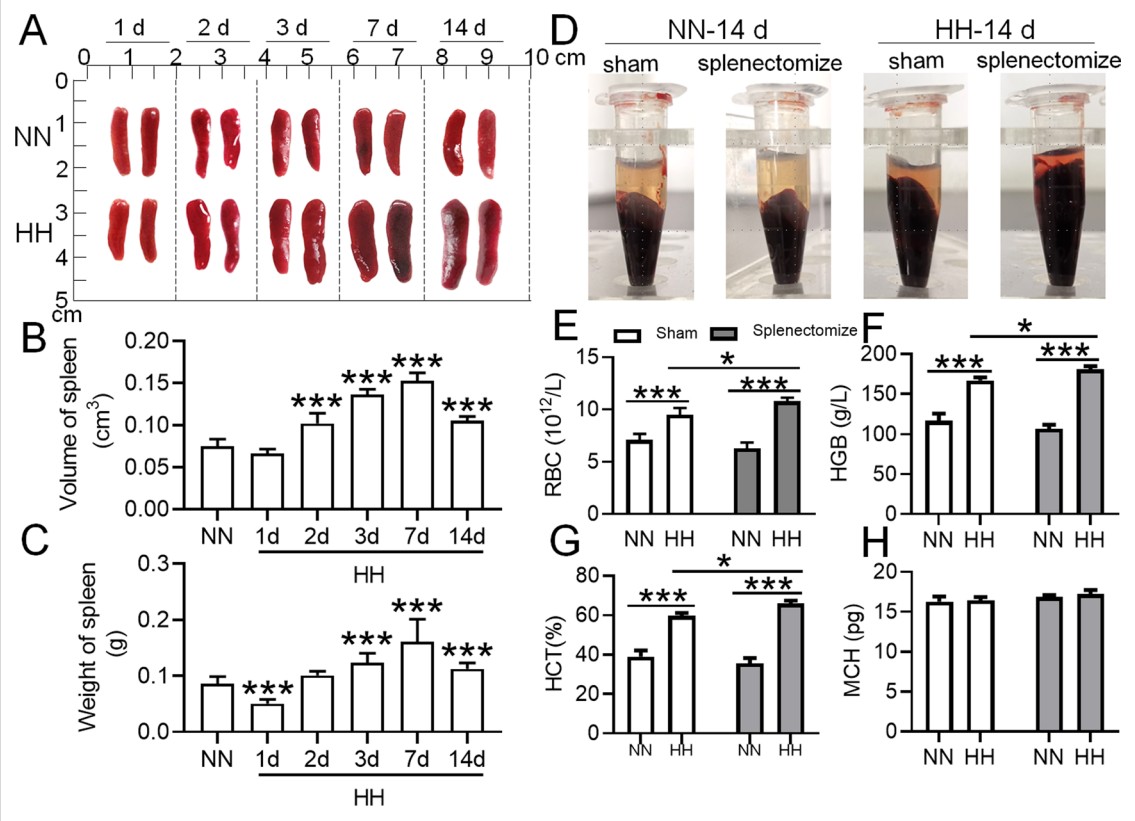

**Figure 2.** The spleen plays an important role in suppressing the immoderate increase in RBCs under HH conditions. C57BL/6 mice with or without splenectomy were treated with NN and HH for varying durations, and the spleen and blood were collected for subsequent analyses. (**A**) Morphological observation, (**B**) Spleen volume, and (**C**) spleen weight was determined. (**D–H**) Blood observation and hematological index detection followed. Data are expressed as the means ± SEM (n=9 per group); * p<0.05, ** p<0.01, *** p<0.001 versus the NN group or the indicated group.

in the spleen from 2 to 14 days after HH exposure. In our research, we also examined the influence of the spleen on RBCs homeostasis under HH conditions. We investigated whether the role of spleen in RBCs clearance under HH conditions could be compensated by the liver or other components of the mononuclear macrophage system. To conduct this, we performed splenectomies on mice and subsequently exposed them to HH conditions for 14 days. This allowed us to monitor RBCs counts and blood deposition. Our findings indicated that, in comparison to both the splenectomized mice under NN conditions and the sham-operated mice exposed to HH, erythrocyte deposition (*Figure 2D*) and counts (*Figure 2E*), as well as HGB (*Figure 2F*) and HCT (*Figure 2G*) levels, significantly increased 14 days post-splenectomy under HH conditions. Meanwhile, MCH levels remained stable (*Figure 2H*). These indices did not vary in mice, regardless of whether they had undergone a splenectomy, under NN conditions (*Figure 2D and E*). These results indicate that in the splenectomized group under NN conditions, erythrophagocytosis is substantially compensated for by functional macrophages in other tissues. However, under HH conditions, our data also suggest that the spleen plays an important role in managing erythrocyte turnover, as indicated by the significant impact of splenectomy on erythrophagocytosis and subsequent RBCs dynamics.

## HH exposure leads to a decrease in splenic macrophages

Considering macrophages as the primary cell type responsible for processing RBCs within the spleen under physiological conditions, we subsequently investigated the population and activity of these macrophages after exposure to HH for 7 or 14 days, employing flow cytometry and single-cell sequencing techniques. Calcein/PI double staining, examined via flow cytometry, revealed a significant decrease in viable splenic cells and a concomitant increase in dead cells following 7 and 14 days of HH exposure (*Figure 3A–C*). This observation was further substantiated by single-cell sequencing, which elucidated a pronounced reduction in the population of splenic macrophages after 7 days of HH

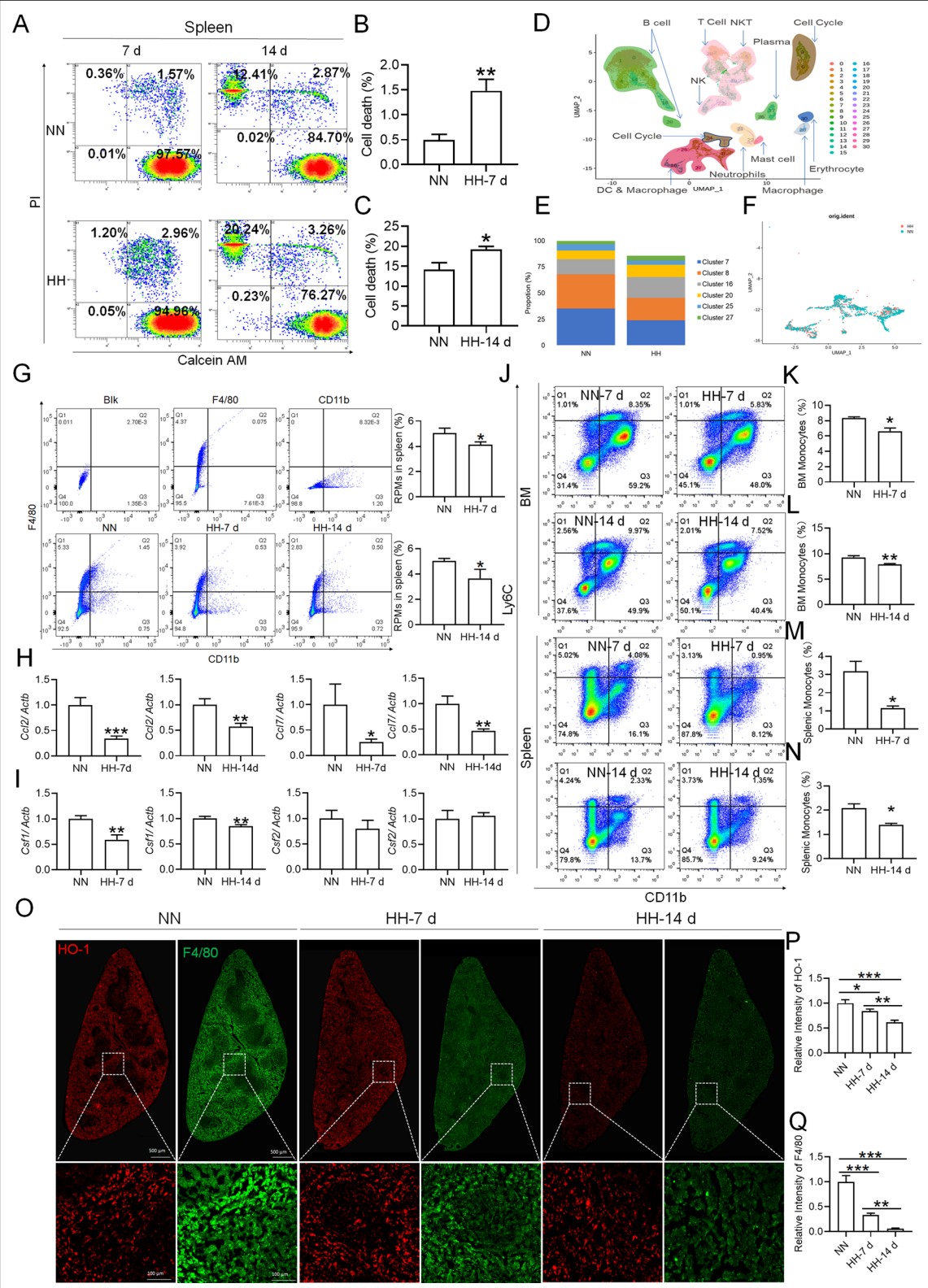

**Figure 3.** HH exposure results in a decrease in the number of splenic macrophages. C57BL/6 mice were treated with NN and HH for 7 or 14 days, and the spleen and blood were collected for subsequent analysis. (**A**) Calcein/PI double staining was analysed by flow cytometry to assess cell viability. (**B and C**) The bar graphs represent the proportions of cell death in total splenic cell population. (**D**) Uniform Manifold Approximation and Projection (UMAP) provided a visualization of spleen cell clusters, with each color representing a unique cluster characterized by specific gene expression profiles.

*Figure 3 continued on next page*

*Figure 3 continued*

(**E**) Comparative analysis of the proportions of distinct macrophage clusters in spleens from NN- and HH-treated mice. (**F**) Presentation of individual macrophages from spleens of mice treated with either NN or HH. (**G**) Analysis of F4/80 and CD11b expression in splenic cells, conducted via flow cytometry. (**H**) qPCR analysis evaluated *Ccl2* and *Ccl7* gene expression in the spleen. (**I**) qPCR analysis of *Csf1* and *Csf2* expression levels in the spleen. (**J**) Flow cytometry facilitated the detection of CD11b and Ly6C double-stained cells in BM and spleen. (**K–L**) Bar graphs represent the proportions of CD11b$^{hi}$Ly6C$^{hi}$ cells in the BM, while (**M–N**) delineate those in the spleen. (**O**) HO-1 and F4/80 expression levels in the spleen were monitored after 0 (NN), 7, and 14 days of HH exposure. (**P**) The relative fluorescence intensities of HO-1 and (**Q**) F4/80, as outlined in (**O**), were quantitatively assessed. Data are expressed as means ± SEM for each group (n=3 per group). Statistical significance is indicated by * p<0.05, ** p<0.01, *** p<0.001 when compared to the NN group or the indicated group.

The online version of this article includes the following figure supplement(s) for figure 3:

**Figure supplement 1.** HH exposure induces RPMs reduction in the spleen.

**Figure supplement 2.** HH exposure induces transformation of RPMs subsets in the spleen.

exposure (*Figure 3D–F*). Additionally, flow cytometry results indicated a marked decrease in the population of RPMs, identified as F4/80$^{hi}$CD11b$^{lo}$, in the spleen post 7 days of HH exposure (*Figure 3G*). Complementary to these findings, immunofluorescence detection demonstrated a significant diminution in RPMs, characterized by F4/80$^{hi}$CD11b$^{lo}$ and F4/80$^{hi}$CD68$^{hi}$ cell populations, after both 7 and 14 days of HH exposure (*Figure 3—figure supplement 1*). Furthermore, our single-cell sequencing analysis of the spleen under NN conditions revealed a predominant association of RPMs with Cluster 0. Intriguingly, HH exposure led to a notable reduction in the abundance of RPMs within this cluster. Pseudo-time series analysis provided insights into the transitional dynamics of spleen RPMs, indicating a shift from Cluster 2 and Cluster 1 towards Cluster 0 under NN conditions. However, this pattern was altered under HH exposure, where a shift from Cluster 0 and Cluster 1 towards Cluster 2 was observed (*Figure 3—figure supplement 2*).

To further elucidate the migration and differentiation of monocytes from the bone marrow to the spleen, we analysed the expression of chemokines *Ccl2*, *Ccl7*, *Csf1*, and *Csf2* in the spleen using qPCR and determined the number of monocytes in the bone marrow (BM) and spleen via flow cytometry. Our results indicated a significant reduction in the expression of *Ccl2*, *Ccl7* (*Figure 3H*), *Csf1*, and *Csf2* (*Figure 3I*) in the spleen after 7 and 14 days of HH exposure. Furthermore, the number of monocytes (Ly6C$^+$/CD11b$^+$) in the bone marrow (*Figure 3J–L*) and spleen (*Figure 3J and M–N*) also exhibited a decline after 7 and 14 days of HH exposure. We evaluated the depletion of macrophages in the spleen under HH conditions by examining the expression and distribution of HO-1 and F4/80. There was a decrease in both HO-1 and F4/80, predominantly within the splenic red pulp following HH exposure (*Figure 3O–Q*). Together, these findings indicate a reduction in the number of splenic macrophages after HH exposure, which could impair the spleen's capacity to process erythrocytes.

## HH exposure suppresses erythrophagocytosis of RPMs in spleen

We investigated the influence of HH exposure on erythrocyte phagocytosis and heme iron recycling within splenic macrophages. To this end, we implemented a dual approach: administering NHS-biotin intravenously to mice, followed by HH exposure, and subsequently introducing PKH67-labelled RBCs into the mice, also followed by HH treatment in vivo. This methodology allowed us to monitor the clearance of RBCs by tracking the retention of both biotin-labelled and PKH67-labelled RBCs using flow cytometry in the blood and spleen of the HH-treated mice. Our findings (as detailed in *Figure 4*) indicated a pronounced decrease in the presence of both biotin-labelled and PKH67-labelled RBCs in the blood and spleen at 7- and 14 day intervals post HH exposure. Notably, while a natural decline in labelled RBCs over time was expected, the rate of decay was significantly more acute in the NN exposure group as compared to the HH group. This observation was consistent across both blood (*Figure 4A and C*; *Figure 4E and G*) and spleen samples (*Figure 4B and D*; *Figure 4F and H*). Furthermore, our analysis revealed that the phagocytic capacity of mouse spleen macrophages toward RBCs was notably diminished following HH exposure, particularly on the 14th day (*Figure 4A–D*; *Figure 4E–H*). To confirm these findings, we conducted additional assessments of the PKH67-positive RPMs cell population through both flow cytometry and immunofluorescence detection in the spleen post HH exposure. The findings revealed that both the population of splenic RPMs (F4/80$^{hi}$CD11b$^{lo}$; *Figure 5A–B*) and the PKH67-positive macrophages (*Figure 5A and C; D-E*) consistently demonstrated

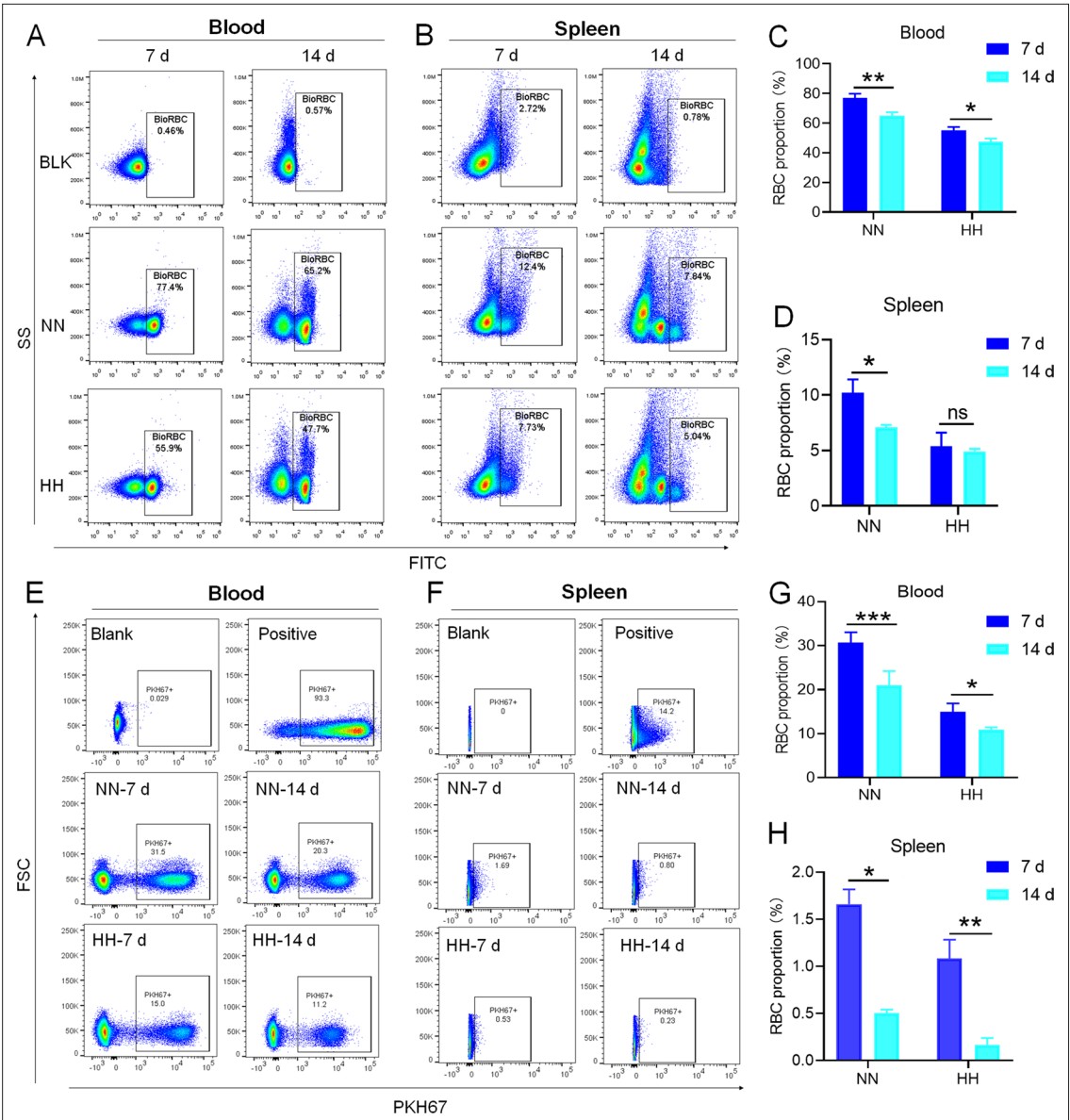

**Figure 4.** HH exposure decreases RBCs clearance both in the blood and spleen. (**A and B**) Flow cytometry measurements illustrated the proportions of FITC-stained RBCs in blood and spleen, respectively, after 7 and 14 days of HH exposure. (**C and D**) Bar graphs depicted the quantified proportions of FITC-positive RBCs in both blood and spleen, respectively. Further, (**E and F**) presented flow cytometry assessments of the proportions of PKH67-labelled RBCs in blood and spleen following 7 and 14 days of HH exposure. The corresponding proportions of PKH67-positive RBCs in blood (**G**) and spleen (**H**) were also shown in bar graph format. The data are expressed as means ± SEM for each experimental group (n=3 per group). Statistical significance is denoted with * p<0.05, ** p<0.01, *** p<0.001, relative to the NN group or as specified in the graph legends. This compilation of data underlines the impact of HH exposure on the reduction of RBCs clearance in both blood and splenic compartments.

a substantial reduction after 7 or 14 days of HH exposure, further reinforcing the impact of HH on the erythrophagocytic function of splenic macrophages.

## Reduced erythrophagocytosis leads to RBCs retention in the spleen under HH exposure

Based on the reduced RPMs and erythrophagocytosis caused by HH exposure in spleen, we next investigated whether RBCs were retained in spleen after HH exposure. Initial examination involved detecting RBCs in the spleen post HH exposure, both with and without perfusion. HE staining (*Figure 6A–B*) and Band 3 immunostaining in situ (*Figure 6C–D and G–H*) revealed a significant increase in RBCs

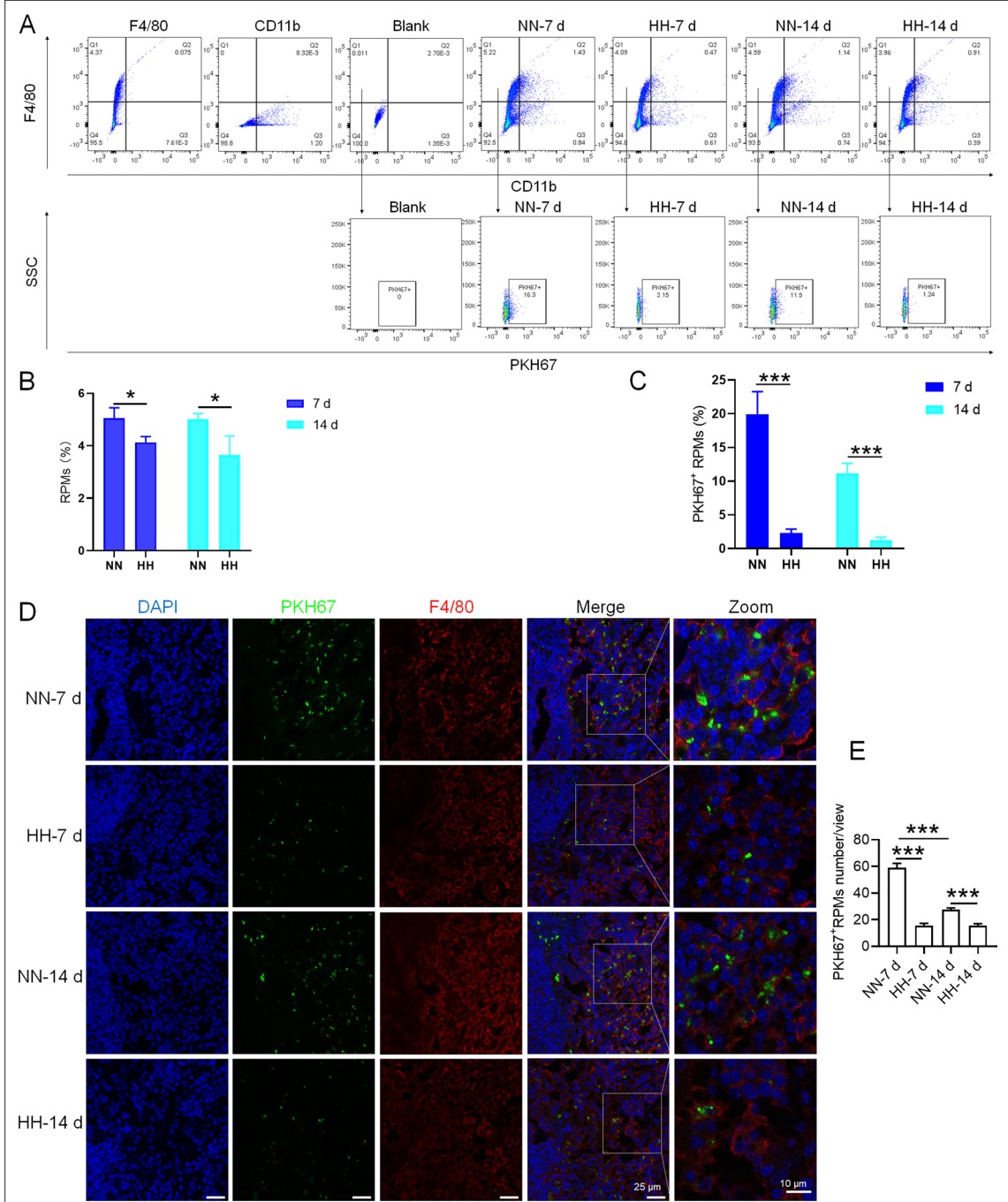

**Figure 5.** HH exposure reduces erythrophagocytosis in the spleen. Mice were administered PKH67-labeled RBCs, followed by exposure to HH for durations of 7 and 14 days. (**A**) Flow cytometry was employed to analyze the proportions of RPMs (identified as F4/80+CD11b-) and PKH67-positive RPMs in the spleen post-exposure. (**B and C**) Bar graphs represent the quantified proportions of RPMs (F4/80+CD11b-) and PKH67-positive RPMs in the spleen, respectively. (**D**) The spleens were subjected to immunofluorescence analysis to detect F4/80 in conjunction with PKH67 fluorescence post-perfusion at 7 and 14 days following HH exposure. (**E**) Subsequent quantification of the number of PKH67-positive F4/80hi cells in the spleen, as depicted in (**D**), was conducted. The data derived from these analyses are expressed as means ± SEM for each group (n=3 per group). Statistical significance is denoted with * p<0.05, and *** p<0.001, relative to the indicated group in the graph legends.

numbers in the spleen following 7 and 14 days of HH exposure. Furthermore, we quantified RBCs retention by employing Wright-Giemsa composite staining on single splenic cells post-perfusion at both 7- and 14 days post HH exposure. The results consistently indicated a substantial elevation in RBC counts within the spleen (*Figure 6E–F*). To enhance the specificity of our investigation, we

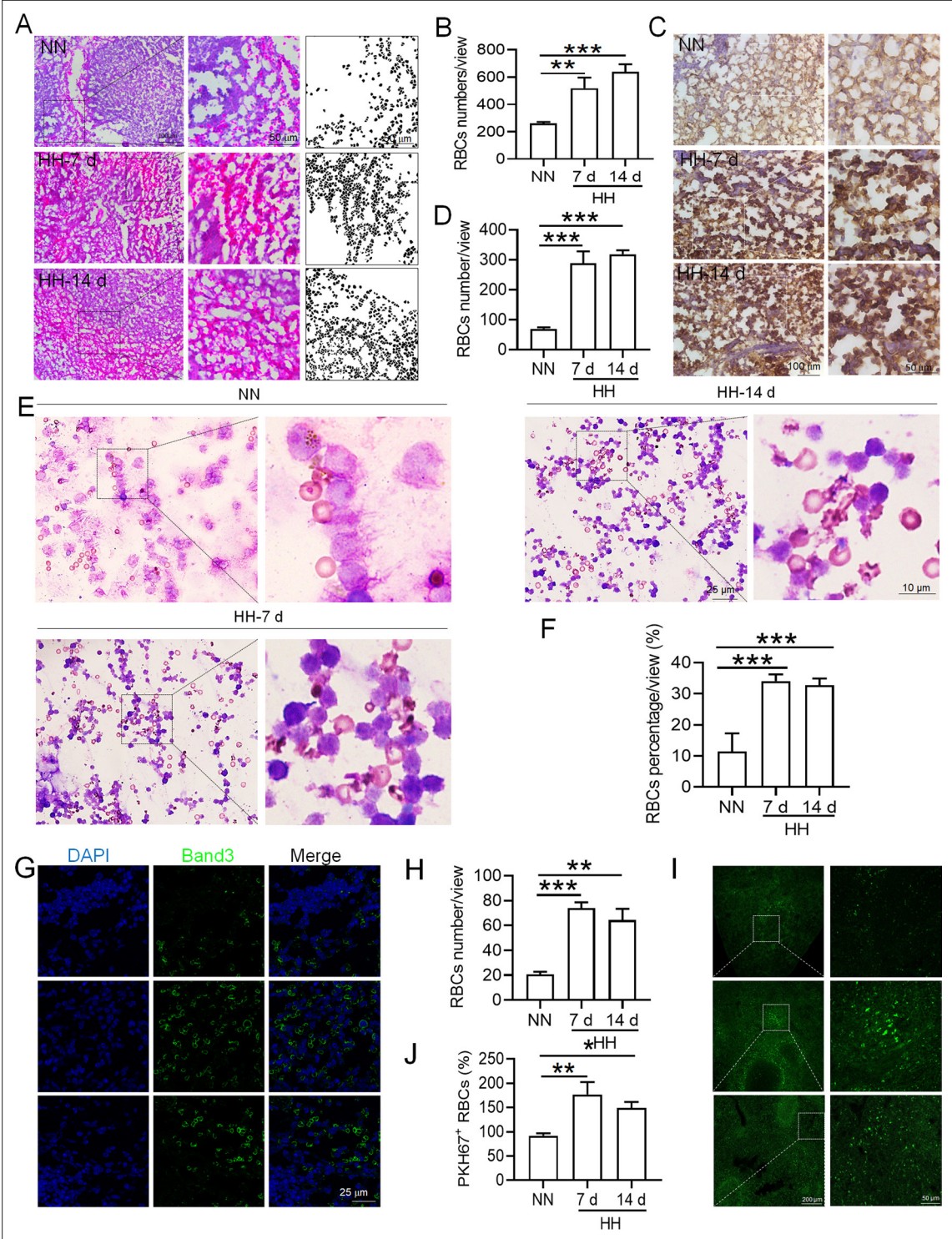

**Figure 6.** HH exposure increases RBCs retention in the spleen. (**A**) HE staining was utilized to examine the presence of RBCs in the spleen post-perfusion at 7 and 14 days following HH exposure. (**B**) The number of RBCs present in the spleen, as observed in (**A**), was quantitatively assessed. (**C and G**) Immunohistochemical and immunofluorescent analyses using Band 3 staining were conducted to evaluate RBCs content in the spleen post-perfusion at 7 and 14 days of HH exposure. (**D and H**) The expression levels of Band 3 in the spleen, as shown in (**C and G**), were quantified. (**E**) Wright-Giemsa composite staining was performed on splenic cells following HH exposure for durations of 0 (NN), 7, and 14 days. (**F**) The proportion of RBCs within the perfused splenic cell population was determined. (**I**) Detection of PKH67 fluorescence in the spleen, without perfusion, was conducted after administering PKH67-labeled RBCs and subjecting them to HH exposure for 7 and 14 days. (**J**) The number of PKH67-positive RBCs within the spleen, as outlined in (**I**), was quantified. Data presented in this figure are expressed as means ± SEM for each experimental group (n=3 per group). Statistical significance is indicated by * p<0.05, ** p<0.01, *** p<0.001 when compared to the NN group or as specified.

labelled RBCs in vitro with PKH67 and then administered them to mice. Following HH exposure for 7 and 14 days, spleen sections were analyzed without perfusion to detect retained PKH67-labeled RBCs. Fluorescence detection techniques was utilized, revealing a marked increase in PKH67-labeled RBCs post HH exposure (*Figure 6I–J*). These results collectively confirm a pronounced impairment in erythrophagocytosis within the spleen under HH conditions. This is evidenced by the observed increase in RBCs deformation and retention following 7 and 14 days of HH exposure. The data thus suggest that HH exposure leads to an increase in RBCs retention in the spleen, highlighting significant alterations in splenic function and RBCs dynamics under hypoxic conditions.

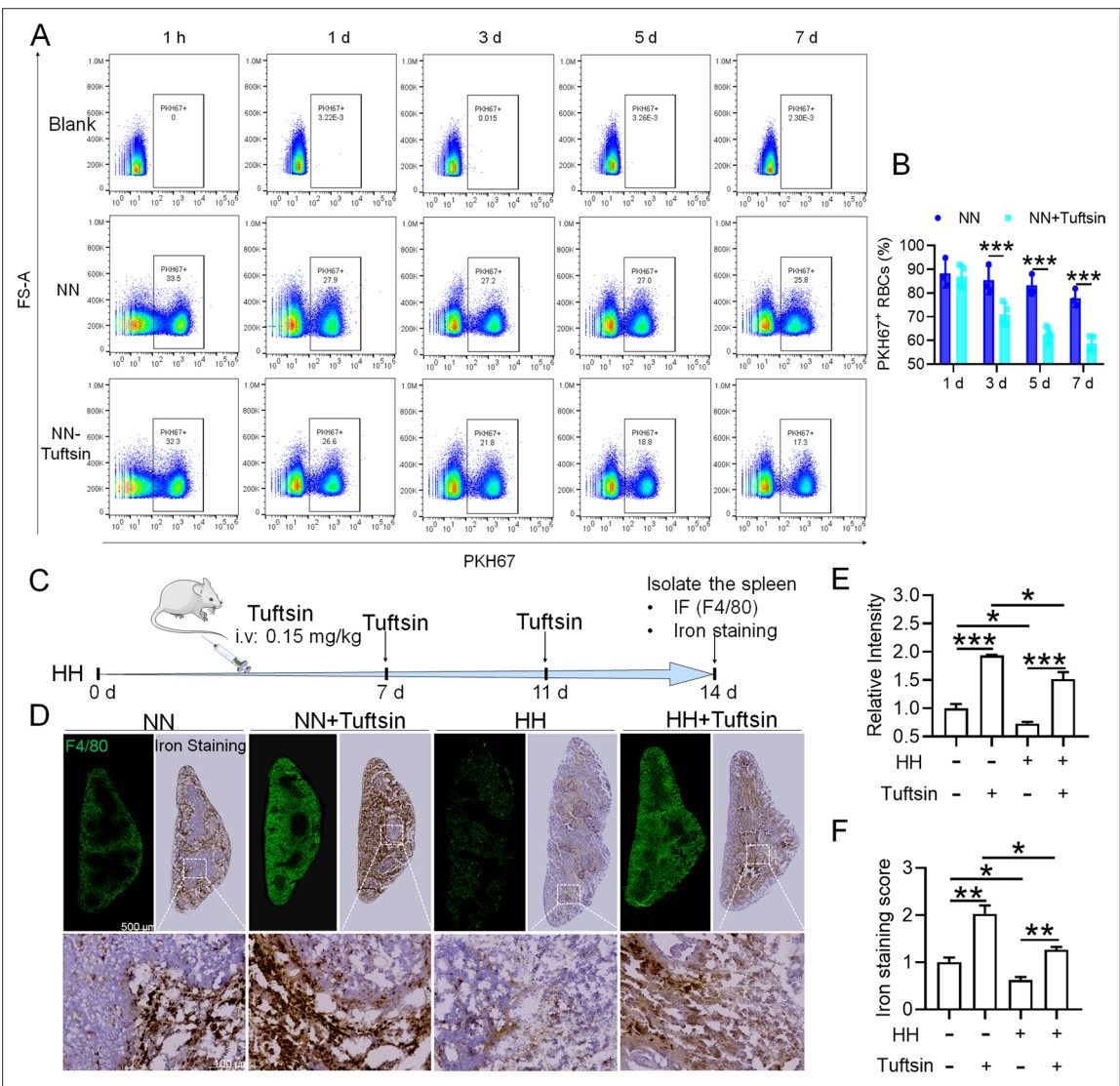

**Figure 7.** Impaired erythrophagocytosis of RBCs after HH exposure leads to a decrease in iron processing capacity in the spleen. To stimulate phagocytosis of macrophages, Tuftsin was administered immediately following the injection of PKH67-labeled RBCs into mice. Subsequent to various exposure durations under NN conditions (1 hr, 1, 3, 5, or 7 days), (**A**) PKH67 fluorescence within the spleen was measured using flow cytometry. (**B**) The percentages of PKH67-positive RBCs in the spleen are represented in bar graph format. (**C**) The experimental design is depicted, wherein C57BL/6 mice (n=3 per group) were administered a single intravenous dose of Tuftsin (0.15 mg/kg) on days 7 and 11, followed by HH exposure until day 14, culminating in spleen isolation for F4/80 immunohistochemistry and iron staining. (**D**) Demonstrates F4/80 and iron staining in the spleen after 14 days of HH exposure with Tuftsin treatment. (**E**) Quantitative analysis of the relative fluorescence intensity of F4/80 expression in the spleen as described in (**D**). (**F**) Semi-quantitative assessment of iron levels in the spleen as outlined in (**D**). Data presented in this figure are articulated as means ± SEM for each group (n=3 per group). Statistical significance is denoted by * p<0.05, ** p<0.01, *** p<0.001 when compared to the NN group or as indicated.

## HH induced reduction in erythrophagocytosis leads to a decrease in the capacity of iron processing in the spleen

To investigate the hypothesis that increased retention of RBCs in the spleen due to HH exposure is a result of impaired erythrophagocytosis by RPMs, we employed a series of experiments. Initially, RBCs were labelled with PKH67 cell linker and then injected into mice. Subsequently, Tuftsin was administered to stimulate the phagocytic activity of macrophages. After 1 hr, 1 day, 3 days, 5 days, or 7 days of NN exposure, PKH67 fluorescence was analyzed via flow cytometry. The findings demonstrated a gradual reduction in PKH67 fluorescence over the course of NN exposure; however, a significant decrease in PKH67 fluorescence was noted in the spleen following Tuftsin administration compared to the NN group. This observation suggested that RBC lifespan is diminished following enhanced erythrophagocytosis (as presented in *Figure 7A–B*). Further investigation was conducted on the ability of the spleen to regulate heme iron recycling under HH conditions through immunofluorescence and iron staining (*Figure 7C*). The results, as illustrated in *Figure 7D*, indicated that both F4/80 expression and iron deposition in the red pulp of the spleen were notably reduced following HH exposure, compared to the NN group. Contrastingly, Tuftsin administration under HH conditions resulted in an upsurge in F4/80 expression and iron deposition in the red pulp (*Figure 7E–F*). These findings collectively suggest that the splenic capacity for erythrocyte phagocytosis and subsequent heme iron recycling is significantly compromised under HH conditions, primarily due to a reduction in RPMs. This reduced erythrophagocytic capacity of the spleen under HH exposure is a critical factor contributing to the decreased efficiency of iron processing and increased RBCs retention in this organ.

## HH exposure induces iron mobilization and ferroptosis in the spleen

To elucidate the precise mechanisms underlying the macrophage reduction caused by HH, we examined the expression of proteins related to iron metabolism and ferroptosis in mice treated with HH and analyzed the corresponding gene expression in peripheral blood mononuclear cells (PBMCs) from healthy humans acutely exposed to HA conditions using GEO data (No. GSE46480). The results showed that, compared to the NN group, the expression levels of HO-1, Ft-L, Ft-H, NCOA4, and xCT were decreased, while Fpn, TfR, and ACSL4 expressions were significantly increased after 7 and 14 days of HH exposure (*Figure 8A–B and D–E*; *Figure 3—figure supplement 1A–D*). Apart from NCOA4, the alterations in gene expressions related to iron metabolism and ferroptosis in PBMCs were consistent with our Western blot results (*Figure 8C and F*). The GPX4 gene and protein expressions remained largely unchanged in both human PBMCs and mouse spleens. The changes in iron metabolism- and ferroptosis-related genes in PBMCs reflect the protein changes in the spleen under HH exposure. We detected $Fe^{2+}$ and lipid ROS levels in the spleen by flow cytometry, and quantified MDA, GSH, and Cys levels using biochemical detection kits. As depicted in *Figure 8*, G and H; *Figure 3—figure supplement 1I-K*, the $Fe^{2+}$ (*Figure 8G*; *Figure 3—figure supplement 1E and F*) and lipid ROS (*Figure 8H*; *Figure 3—figure supplement 1G and H*) levels in the spleen increased significantly after HH exposure. Additionally, the MDA content (*Figure 8I*; *Figure 3—figure supplement 1I*) in the spleen increased, whereas Cys (*Figure 8J*; *Figure 3—figure supplement 1J*) and GSH (*Figure 8K*; *Figure 3—figure supplement 1K*) levels significantly decreased after HH exposure. To ascertain the increased $Fe^{2+}$ primarily emanated from the red pulp, we detected $Fe^{2+}$ deposition and distribution in the spleen by Lillie stain following 7 and 14 days of HH exposure (*Figure 8L–M*; *Figure 3—figure supplement 1L–M*). These results demonstrated increased $Fe^{2+}$ primarily deposited in the red pulp of the spleen. Taken together, these findings suggest that iron mobilization in the spleen was enhanced and ferroptosis was induced after 7 days of HH exposure.

## Hypoxia induces ferroptosis in primary splenic macrophages

To investigate whether hypoxia exposure can trigger ferroptosis in macrophages, we measured alterations in $Fe^{2+}$ and lipid ROS levels, cell viability, phagocytosis, and ferroptosis-related protein expressions in primary splenic macrophages under hypoxia in the presence of a ferroptosis inhibitor (Fer-1). As depicted in *Figure 9*, compared with the normoxia control group (Nor), $Fe^{2+}$ (*Figure 9A–C*) and lipid ROS (*Figure 9D–E*) levels significantly increased, whereas the number of viable cells (*Figure 9G–H*) and phagocytosis ability (*Figure 9I*) significantly decreased after 24 h of hypoxia (Hyp) treatment. In addition, prolonged hypoxia exposure led to a gradual increase in ACSL4 expression, while xCT and GPX4 expressions decreased (*Figure 9F*). Fer-1 mitigated the increase in $Fe^{2+}$ content

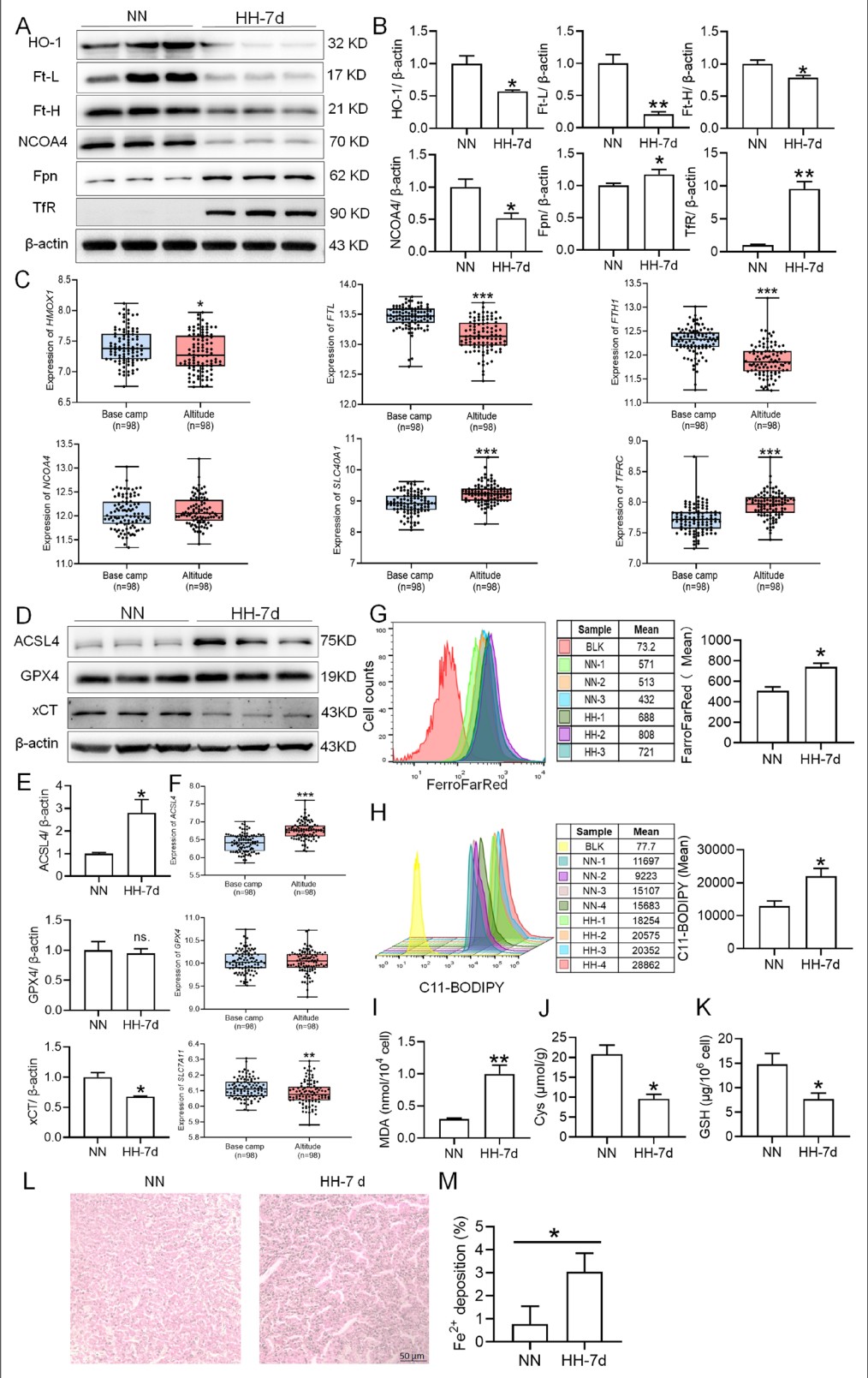

**Figure 8.** HH exposure enhances iron mobilization and induces ferroptosis in the spleen. The C57BL/6 mice were treated with NN and HH for 7 days, and the spleen was collected for subsequent detection. (**A**) Western blot detection of HO-1, Ft-L, Ft-H, NCOA4, Fpn and TfR protein expression in spleen. (**B**) Statistical analysis of HO-1, Ft-L, Ft-H, NCOA4, Fpn, and TfR protein expression. (**C**) GEO data analysis of HMOX1, FTL, FTH1, NCOA4,

*Figure 8 continued on next page*

*Figure 8 continued*

SLC40A1, and TFRC mRNA expression in PMBCs before and after climbing to HA (n=98). (**D**) Western blot analysis for ACSL4, GPX4, and xCT protein expression in the spleen, with (**E**) depicting the statistical analysis of these protein levels. (**F**) GEO data analysis of ACSL4, GPX4 and SLC7A11 mRNA expression in PMBCs before and after reaching HA (n=98). (**G**) The content of $Fe^{2+}$ in the spleen was detected using the FerroFarRed probe by flow cytometry. (**H**) The level of lipid ROS in the spleen was detected using the C11-BODIPY probe by flow cytometry. The MDA (**I**), Cys (**J**) and GSH (**K**) levels in the spleen were detected using biochemical detection kits, respectively. (**L**) Lillie staining of $Fe^{2+}$ in the spleen after 7 days of HH exposure. (**M**) $Fe^{2+}$ deposition in the spleen as described in (**L**) was quantified. Data are expressed as the means ± SEM (n=3 per group); * p<0.05, ** p<0.01, *** p<0.001 versus the NN group or the indicated group.

The online version of this article includes the following source data and figure supplement(s) for figure 8:

**Source data 1.** PDF containing *Figure 8A* and original scans of the relevant western blot analysis (anti-HO-1, anti-Ft-L, anti-Ft-H, anti-NCOA4, anti-Fpn, anti-TfR and anti-β-actin) with highlighted bands and sample labels.

**Source data 2.** PDF containing *Figure 8D* and original scans of the relevant western blot analysis (anti-ACSL4, anti-GPX4, anti-xCT and anti-β-actin) with highlighted bands and sample labels.

**Figure supplement 1.** HH exposure promotes iron mobilization and induces ferroptosis in the spleen.

**Figure supplement 1—source data 1.** PDF containing *Figure 8—figure supplement 1A* and original scans of the relevant Western blot analysis (anti-HO-1, anti-Ft-L, anti-Ft-H, anti-NCOA4, anti-Fpn, anti-TfR and anti-β-actin) with highlighted bands and sample labels.

**Figure supplement 1—source data 2.** PDF containing *Figure 8—figure supplement 1C* and original scans of the relevant western blot analysis (anti-ACSL4, anti-Gpx4, anti-XCT, and anti-β-actin) with highlighted bands and sample labels.

---

(*Figure 9J–K*) and reversed the expression changes in ACSL4, xCT, and GPX4 induced by hypoxia exposure (*Figure 9L–O*). Simultaneously, Fer-1 reversed the alterations in MDA, Cys, and GSH levels induced by hypoxia (*Figure 9P-R*). These findings confirm that hypoxia induced ferroptosis in primary splenic macrophages.

## Discussion

The purpose of this study was to explore the effects and mechanisms of HA/HH exposure on erythrophagocytosis and iron circulation in mouse spleens. Here, we used an HH chamber to simulate 6000 m HA exposure and found that HH exposure induced iron mobilization and activated ferroptosis in the spleen, especially in RPMs, which subsequently inhibited the phagocytosis and clearance of RBCs. Finally, chronic exposure to HA/HH may promote the retention of RBCs in the spleen, cause splenomegaly, advance RBC production, and promote the occurrence and development of HAPC.

RBCs/erythrocytes are the carriers of oxygen and the most abundant cell type in the body. Erythrocytes are rapidly increased by triggering splenic contraction in acute HA/HH exposure (*Schagatay et al., 2020*) and stimulating erythropoiesis in subsequent continuous or chronic HA/HH exposure. Changes in spleen morphology and size are closely related to the spleen's ability to recover RBCs (*Khairullah and Jackson, 2021*), and studies have shown that the spleen is in a contraction state after short-term exposure to HA/HH, and approximately 40% of the increase in RBCs is due to the contraction of the spleen (*Alafi and Cook, 1956*; *Schagatay et al., 2020*; *Richardson et al., 2008*). In the present study, mice were treated under HH to mimic HA exposure (mainly mimicking the low oxygen partial pressure environment of 6000 m HA) for different times according to other studies (*Lin et al., 2011*; *Brent et al., 2022*). Our results showed that HH exposure significantly increased the number of RBCs and HGB contents. However, short-term (1 day) HH exposure caused spleen contraction and induced splenomegaly after 3 days of HH treatment. This contraction can trigger an immediate release of RBCs into the bloodstream in instances of substantial blood loss or significant reduction of RBCs. Moreover, elevated oxygen consumption rates in certain animal species can be partially attributed to splenic contractions, which augment haematocrit levels and the overall volume of circulating blood, thereby enhancing venous return and oxygen delivery (*Dane et al., 2006*; *Longhurst et al., 1986*). Thus, we hypothesized that the body, under such conditions, is incapable of generating sufficient RBCs promptly enough to facilitate enhanced oxygen delivery. Consequently, the spleen reacts by releasing its stored RBCs through splenic constriction, leading to a measurable reduction in

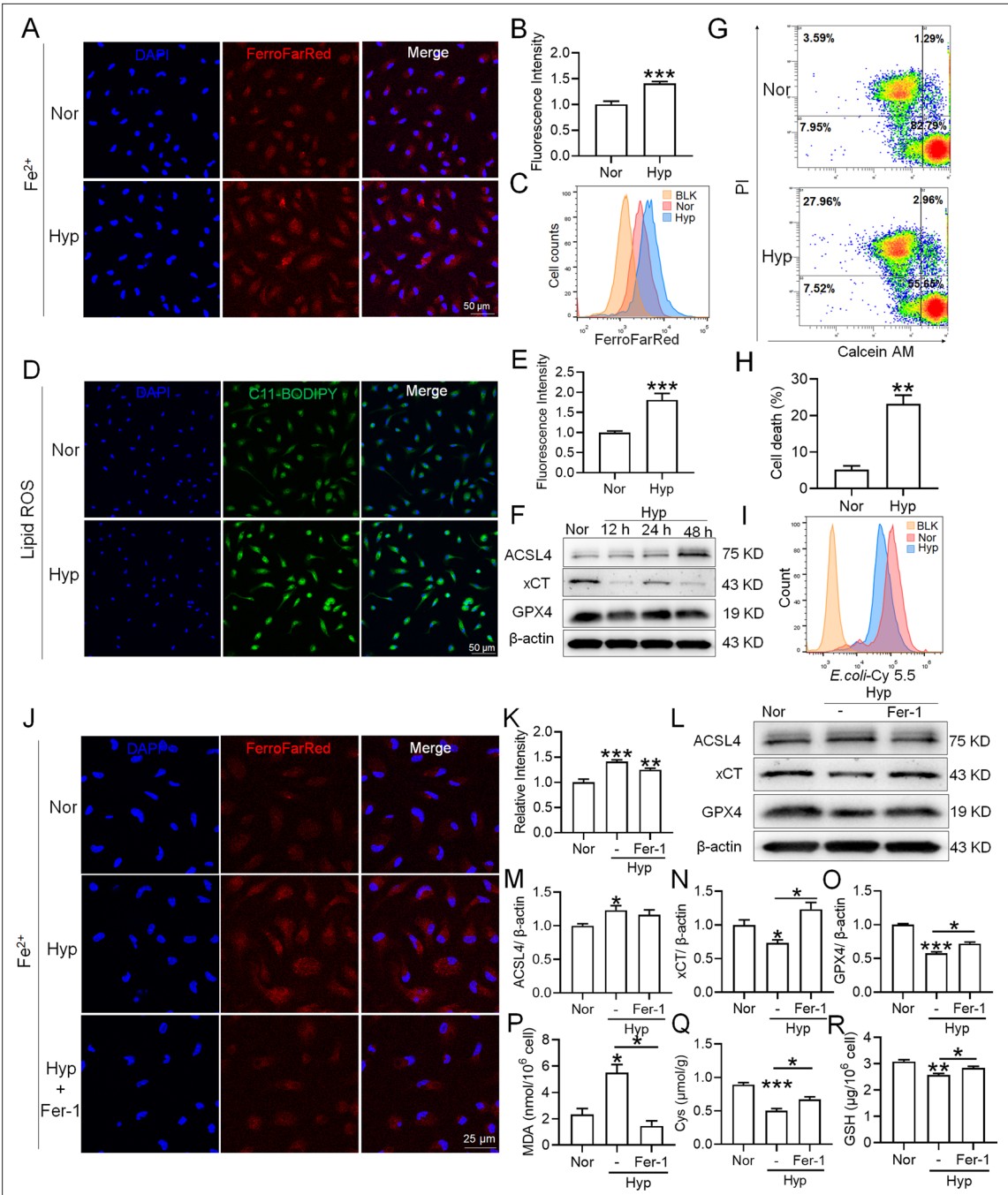

**Figure 9.** Hypoxia enhances the ferroptosis in splenic macrophage. Splenic macrophages were cultured under 1% hypoxia for varying durations (0, 12, 24, 48 hr), either with or without a pre-treatment of 10 μM Fer-1 for 1 hr, before being collected for further examination. (**A**) Immunofluorescence detection of Fe²⁺ levels in macrophages after hypoxia exposure for 24 hr using the FerroFarRed probe under a fluorescence microscope. (**B**) Quantitative results of fluorescence intensity in the macrophages described in A. (**C**) Flow cytometry detection of Fe²⁺ levels in macrophages exposed to hypoxia for 24 hr using the FerroFarRed probe. (**D**) Immunofluorescence detection of lipid ROS levels in macrophages exposed to hypoxia for 24 hr using the C11-BODIPY probe under a fluorescence microscope. (**E**) Quantitative results of mean fluorescence intensity in the macrophages described in (**D**). (**F**) Western blot detection of ACSL4, GPX4, and xCT protein expression in macrophages under hypoxia for different times. (**G**) Flow cytometry was employed to analyze Calcein/PI double staining in macrophages following a 24 hr hypoxia exposure. (**H**) Cell death proportions in macrophages are given as bar graphs. (**I**) Macrophage phagocytic activity against Cy5.5-labelled *E. coli* following 24 hr of hypoxia exposure was assessed using flow cytometry in vitro. (**J**) Fe²⁺ levels in macrophages pre-treated with Fer-1, then exposed to 24 hr of hypoxia, were detected via the FerroFarRed probe in immunofluorescence. (**K**) Quantitative results of fluorescence intensity in the macrophages described in (**J**). (**L**) Western blot detection of ACSL4, GPX4, and xCT protein expression in macrophages pre-treated with Fer-1, then followed by 24 hr of hypoxia exposure. (**M–O**) Statistical analysis of ACSL4, xCT,

*Figure 9 continued on next page*

*Figure 9 continued*

and GPX4 protein expression in L. MDA (**P**), Cys (**Q**), and GSH (**R**) levels in macrophages pre-treated with Fer-1 and subsequently exposed to 24 hr of hypoxia were detected using kits and biochemical methods. Data are expressed as the means ± SEM (n=3 per group); * p<0.05, ** p<0.01, *** p<0.001 versus the NN group or the indicated group.

The online version of this article includes the following source data for figure 9:

**Source data 1.** PDF containing *Figure 9F* and original scans of the relevant western blot analysis (anti-ACSL4, anti-xCT, anti-GPX4, and anti-β-actin) with highlighted bands and sample labels.

**Source data 2.** PDF containing *Figure 9L* and original scans of the relevant western blot analysis (anti-ACSL4, anti-xCT, anti-GPX4, and anti-β-actin) with highlighted bands and sample labels.

spleen size. These results are consistent with other research results; that is, HH or HA exposure affects spleen morphology and further affects RBCs counts and HGB levels (*Holmström et al., 2021*; *Holmström et al., 2020*; *Pernett et al., 2021*).

HAPC is a common chronic HA disease characterized by excessive proliferation of RBCs caused by HH conditions (*Wang et al., 2023*). Under physiological conditions, RBC homeostasis is maintained by balancing erythropoiesis and erythrocyte clearance (*Dzierzak and Philipsen, 2013*). The ability of RBC disposal in the spleen for iron recycling under HA/HH exposure was important to RBC regeneration in BM (*Pivkin et al., 2016*). Thus, we hypothesized that erythrophagocytosis and iron recycling in the spleen were altered by HA/HH exposure and further disturbed RBC homeostasis and affected the progression of HAPC. We found that compared with sham group mice, HH significantly increased the contents of RBCs and HGB in the blood of splenectomy mice. This strongly verified that the spleen is a key organism that maintains RBC magnitude within a certain range of physiological statuses under HA/HH conditions.

As originally proposed by Metchnikoff in the 19th century, macrophages, especially RPMs, play a pivotal role in the regulation of RBCs and iron homeostasis (*Wynn et al., 2013*; *Gammella et al., 2014*). We detected the macrophage population in the spleen and found that HH exposure for 7 and 14 days reduced the total macrophages in the spleen. We next observed whether the migration and differentiation of monocytes from the BM to the spleen supplemented the reduced macrophages after HH treatment and maintained their homeostasis. It is well known that splenic erythrophagocytosis is a dynamic process (*Dzierzak and Philipsen, 2013*). Upon phagocytosing erythrocytes in macrophages, spleen tissue (including macrophages and fibroblasts) produces the chemokine C–C motif chemokine ligand 2 and 7 (CCL2 and CCL7) to recruit blood monocytes to the spleen (*Zhang et al., 2022*; *Bellomo et al., 2020*). In our study, the expression of *Ccl2*, *Ccl7*, and *Csf1* in the spleen decreased after 7 and 14 days of HH exposure. Furthermore, we found that HH exposure inhibited Ly6C[hi] monocyte migration from the BM to spleen, where these cells subsequently differentiated into RPMs. The combination of decreased splenic RPMs, along with the reduced BM-derived Ly6C[hi] monocytes, suggests that the homeostasis of the RPMs population in circulating monocytes was also inhibited after the depletion of RPMs induced by HH exposure. The observed decrease in monocytes within the BM is likely attributable to the fact that monocytes and precursor cells for RBCs both originate from the same hematopoietic stem cells within the BM (*Bapat et al., 2021*). As such, the differentiation to monocyte is reduced under hypoxic conditions, which may subsequently cause a decrease in migration to spleen.

HO-1 is a pivotal antioxidant and cytoprotective enzyme, instrumental in catalyzing the degradation of heme, thereby maintaining heme homeostasis and mitigating free heme-induced toxicity (*Vijayan et al., 2018*). Notably, it has been posited that mice deficient in HO-1 exhibit a significant depletion of RPMs, emphasizing the important role of this enzyme in the development and maintenance of RPM after RBCs clearance (*Kovtunovych et al., 2010*). In our study, the observed decrease in HO-1 protein levels in the red pulp suggests a potential reduction in the number of macrophages induced by HH exposure. This inference is made despite the general understanding that HO-1 expression is typically upregulated by hypoxia-inducible factor 1 (HIF-1) under hypoxic conditions (*Lee et al., 1997*; *Dawn and Bolli, 2005*). Along with the decrease in the number of macrophages caused by HH, we also noticed that the phagocytic ability of macrophages in the spleen is inhibited, as evidenced both in vitro and in vivo. Furthermore, the administration of tufftsin markedly enhanced heme iron processing in the spleen under HH conditions. However, our findings appear to diverge from those reported by Anand et al., which suggested an increase in macrophage phagocytosis under hypoxic

conditions, mediated by HIF-1α (*Anand et al., 2007*). This discrepancy may stem from the differing methodologies employed; whereas Anand et al. utilized intermittent hypoxia in their research, our study was characterized by consistent hypoxia. However, it should be acknowledged that using HO-1 expression as an alternative marker to quantify macrophage numbers is not without limitations. Variations in HO-1 expression per cell can yield misleading results, and the localization of this effect to the red pulp does not definitively equate to a conclusion applicable to macrophages, given the inherent heterogeneity of this region and the spleen overall. Considering the multifaceted nature of spleen function and cellular dynamics under hypoxic stress, this complexity requires careful interpretation of our data.

Macrophages play a crucial role in maintaining erythrocyte homeostasis, partially due to their function in digesting the HGB content of cleared RBCs and recycling the iron back to erythroid progenitors for heme synthesis and HGB production (*Korolnek and Hamza, 2015*). In mammals, most of the bodily iron exists in the form of heme, with the most substantial pool comprising HGB (*Hamza and Dailey, 2012*). HGB contains four prosthetic hemoglobins, and each mature RBC is thought to contain approximately $1.2 \times 10^9$ heme moieties (*Korolnek and Hamza, 2015*). As previously mentioned, most of the iron required to sustain erythropoiesis is derived from recycled RBCs. In general, RPMs are equipped with molecular machinery capable of neutralizing the toxic effects of heme and metabolizing iron (*Soares and Hamza, 2016*; *Ganz, 2016*). Nonetheless, any disruptions in the process of erythrophagocytosis, the uptake and degradation of erythrocytes by macrophages, could potentially result in abnormal iron metabolism, potentially manifesting as conditions such as anemia or iron overload (*Soares and Hamza, 2016*). The release of heme upon processed RBCs constitutes a permanent and considerable threat for iron cytotoxicity in macrophages and may eventually result in a specific form of programmed cell death termed ferroptosis (*Dixon et al., 2012*), which may cause decreased cell counts (*Zhang et al., 2020*; *Chen et al., 2020*).

Accordingly, we investigated the iron metabolism status and explored ferroptosis in spleen/macrophages after HH/hypoxia treatment and found that HH exposure prompted iron mobilization, especially in ferritinophagy and lipid peroxidation in the spleen. Interestingly, we found that all gene expression changes in PBMCs after acute HA exposure in humans, except for NCOA4, were similar to those in the spleen of mice exposed to HH for 7 and 14 days. This is probably because NCOA4 in the spleen mediates iron release from Ft-L and facilitates iron reuse, while PBMCs do not possess this function in vivo. These results not only indicated that HH exposure induces spleen ferroptosis but also implied that the gene expression changes in PBMCs under HA/HH may reflect RBC processing functions in the spleen and further indicate the iron metabolism status in the clinic. We further found that the exact mechanism of macrophage ferroptosis induced by HH exposure was caused by increased $Fe^{2+}$ and decreased antioxidative system expression, which finally resulted in lipid peroxidation of macrophages. It has been proposed that heme catabolism by HO-1 protects macrophages from oxidative stress (*Soares and Hamza, 2016*). The enhanced ROS and lipid peroxidation in vivo were also consistent with the decreased HO-1 expression in the spleen. In addition, 1% hypoxia treatment induced ferroptosis in vitro, which was reduced by treatment with ferrostatin-1, a ferroptosis inhibitor. However, our data were inconsistent with the study of Fuhrmann et al., which reported that hypoxia inhibits ferritinophagy and protects against ferroptosis (*Fuhrmann et al., 2020*). We hypothesized that the enhanced iron demand for new RBCs regeneration in BM caused by HH leads to a decrease in the expression of NCOA4 and Ft-L proteins in the spleen. On the other hand, hypoxia inhibited anti-ferroptosis system expression, including reduced Gpx4 expression and increased lipid ROS production. The results were supported by the group of Youssef LA et al, who reported that increased erythrophagocytosis induces ferroptosis in RPMs in a mouse model of transfusion (*Youssef et al., 2018*). However, as most studies have reported, ferroptosis is not only characterized by increased lipid ROS but also significantly shrinking mitochondria (*Xie et al., 2016*). We never found shrinking mitochondria in RPMs after HH exposure (data not shown). Gao et al. proposed that mitochondria played a crucial role in cysteine deprivation-induced ferroptosis but not in that induced by inhibiting glutathione peroxidase-4 (GPX4), the most downstream component of the ferroptosis pathway (*Gao et al., 2019*). Interestingly, in our in vivo study, Gpx4 expression was also not changed after HH exposure. It is still not clear whether mitochondria are involved in ferroptosis. Whether HH treatment caused mitochondrial swelling directly and the exact mechanism involved in this process still need to be further investigated.

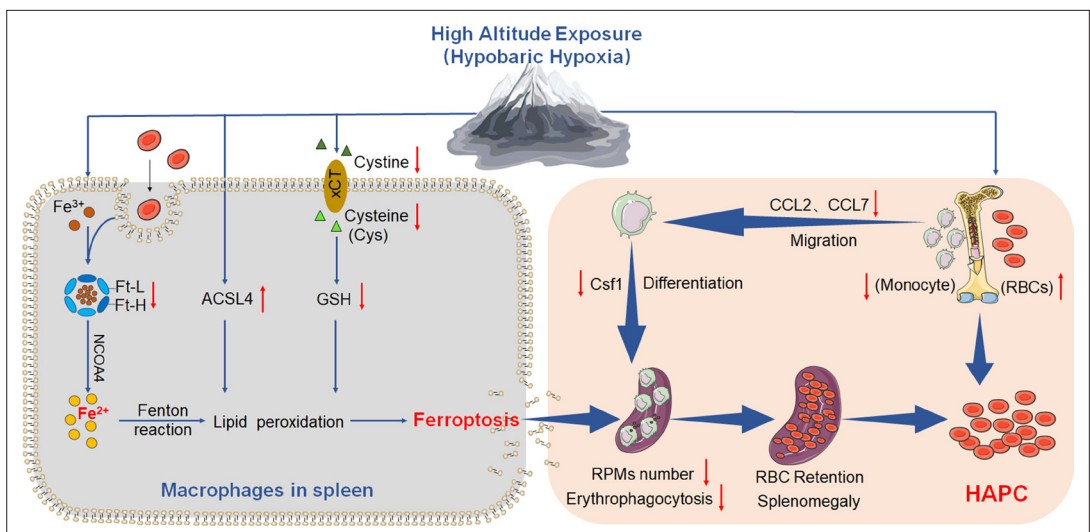

**Figure 10.** HA/HH exposure suppresses erythrophagocytosis by inducing macrophages ferroptosis in the spleen. This figure summarizes the multifaceted impact of HA and HH exposure on splenic macrophage function. HA/HH exposure is demonstrated to trigger significant iron mobilization, notably increasing $Fe^{2+}$ levels, which are further elevated in the spleen and macrophages of mice. Concurrently, HA/HH exposure is associated with an upregulation of ACSL4 expression, coupled with a suppression of xCT expression and GSH production. This alteration results in enhanced lipid peroxidation, culminating in the escalation of macrophage ferroptosis. Additionally, the exposure adversely affects monocyte migration and differentiation from BM to the spleen. Collectively, these effects of HA/HH exposure contribute to a notable inhibition of erythrophagocytosis, thereby promoting erythrocytosis and potentially exacerbating the progression of HAPC. This comprehensive analysis underscores the intricate interplay between environmental factors and cellular mechanisms in the spleen, which could significantly influence the pathogenesis of HAPC.

It is also possible that the spleen may adapt to the increased load of RBCs by expanding the red pulp, leading to splenomegaly (*Li et al., 2018*). The red pulp is highly vascular and contains numerous sinuses lined with macrophages (*Mebius and Kraal, 2005*). This expansion could potentially accommodate a greater number of RBCs, thus enhancing the splenic capacity for erythrophagocytosis and iron recycling. However, these compensatory mechanisms may be insufficient or become overwhelmed over time, leading to pathological conditions such as HAPC. Further research is necessary to fully understand the consequences of splenic ferroptosis on RBC homeostasis and how these effects could be mitigated or reversed. Finally, as these observations are based on experimental models, they should be corroborated in human studies to assess their clinical relevance and potential implications for managing conditions related to high altitude exposure. The potential for therapeutic interventions that target ferroptosis, erythrophagocytosis, and iron recycling should also be explored, as these could provide new avenues for treating HAPC and other conditions related to HA exposure.

Based on the above experiments, we propose the 'spleen theory' of HAPC as depicted in *Figure 10*. It hypothesizes that HH exposure precipitates ferroptosis within the spleen, particularly affecting RPMs. This process, in turn, impedes erythrophagocytosis, RBCs clearance, HGB processing, and iron recycling. Consequently, this leads to an accumulation of RBCs within the spleen, exacerbating splenomegaly and perpetuating the production of RBCs under HH conditions. The 'spleen theory' of HAPC provides a novel perspective on the physiological responses to hypoxia and highlights the role of the spleen as a vital organ in managing erythrocyte homeostasis under high altitude/hypoxic conditions. These findings may be clinically relevant to the pathological conditions of HAPC progression.

## Materials and methods
### HH mouse exposure model
Male C57BL/6 male mice (at 8 weeks of age) were obtained from the animal experimental centre of Nantong University. Mice were adapted to the facilities for 3 days before experiments, and then

randomly divided into an HH exposure group and a control group. Mouse models of HH exposure were established by placing the mice in the decompression chamber (TOW-INT TECH, ProOx-810, China) for 1, 2, 3, 7, and 14 days. The parameters were set as follows: 6000 m above sea level, oxygen partial pressure of 11 kPa, lifting speed of 20 m/s, chamber pressure of 54 kPa, temperature of 25 °C, humidity of 40%, and 12 hr light/dark cycles. The hypobaric chamber was configured to ensure a ventilation rate of 25 air exchanges per minute. Tuftsin (0.15 mg/kg, MCE, HY-P0240, USA) was used to stimulate the phagocytosis of macrophages (*Corazza et al., 1999*), and was injected at 7 and 11 days during the 14 days period of mice exposed to HH. The control group animals were maintained in a similar chamber for the same amount of time under normobaric normoxia (NN) conditions. All measurements were conducted by investigators who were blinded to the assignments of the experimental groups.

## Splenectomy

After anesthesia, the mice were placed in a supine position, and their limbs were fixed. The abdominal operation area was skinned, disinfected, and covered with a sterile towel. A median incision was made in the upper abdomen, followed by laparotomy to locate the spleen. The spleen was then carefully pulled out through the incision. The arterial and venous directions in the splenic pedicle were examined, and two vascular forceps were used to clamp all the tissue in the main cadre of blood vessels below the splenic portal. The splenic pedicle was cut between the forceps to remove the spleen. The end of the proximal hepatic artery was clamped with a vascular clamp, and double or through ligation was performed to secure the site. The abdominal cavity was then cleaned to ensure there was no bleeding at the ligation site, and the incision was closed. Post-operatively, the animals were housed individually. Generally, they were able to feed themselves after recovering from anesthesia and did not require special care.

## Splenic macrophage culture and treatments

After isoflurane anaesthesia, mice were transcardially perfused with precooled PBS, and then splenic tissue was collected. The spleen was then dissected into minuscule fragments, each approximately 1 mm$^3$ in volume, and separated in cold, phenol red-free Hanks' balanced salt solution (HBSS; Gibco, Grand Island, NY, USA). The cell extracts were centrifuged at 500 rpm for 5 min at 4 °C. The pellets were then resuspended and incubated with ammonium-chloride-potassium (ACK) lysing buffer (Thermo Fisher Scientific, USA) for 3 min at room temperature. Next, cell extracts were purified in a four-phase Percoll (Sigma−Aldrich, St. Louis, MO, USA) gradient: 0, 30, 40, and 50% in HBSS. After centrifugation at 500 rpm for 20 min at 4 °C, the macrophage-enriched fraction was collected at the interface between the 30% and 40% phases. Cells were washed in two rounds with HBSS and then resuspended in AIM V medium (Thermo Fisher Scientific). The cells in this medium were cultured at 37 °C and 5% $CO_2$ for 12 hr. After the cells were treated with ferrostatin-1 (Fer-1, 2 μm; Selleck Chemicals, S7243, USA) 1 hr in advance, the cells were placed in a hypoxia workstation (Invivo2, Ruskinn, UK) for 24 hr.

## Blood smear and hematological indices

The collected blood was gently mixed with EDTA (15 g/L) to obtain anticoagulant blood, which was detected by a hematology analyser (XE-5000, Sysmex Corporation, Japan). Blood smear preparation method: a blood sample is dropped onto a slide, and another slide is used to push the blood sample into a uniform thin layer at a constant speed. After the blood samples were dried, Wright stain solution (G1040, Solarbio, China) was added and incubated for 1 min. Then, the slide was rinsed with water. Blood smears were photographed with a DM4000B microscope (Leica, Germany).

## Lillie staining and Hematoxylin and Eosin (HE) staining

The Lillie staining method was employed to ascertain $Fe^{2+}$ deposition within the spleen, according to the protocol delineated by *Liu et al., 2021*. Briefly, paraffin-embedded spleen sections underwent a process of dewaxing and rehydration via xylene and gradient alcohol. Then, the Lillie staining solution (G3320, Solarbio, China) was applied at 37 °C for 50 min in a light-restricted environment, improved by a 5-min PBS wash. Subsequently, a mixture comprising 30% hydrogen peroxide and methanol was incubated for 20 min, and then the sections were subjected to two 5 min washes with PBS. The

DAB developing solution, which is not included in the Lillie bivalent iron staining kit, was applied for light-sensitive staining. The sections were stained with a nuclear fast red solution in a light-restricted setting for 10 min, followed by a 5 second wash with ddH$_2$O. Lastly, the sections were dehydrated with gradient alcohol, cleared with xylene, and mounted with neutral resin. HE staining was employed in the spleen according to our previous study (*Jiang et al., 2022*).

## Wright-Giemsa composite stain

Mice were anaesthetized and perfused with saline. The spleen was subsequently excised to prepare a single cell suspension, achieved by filtering through a 70 μm filter. Samples were centrifuged at 500 × *g* for 5 min for cell separation. The cell pellet was subjected to three PBS washes. The suspension was then smeared onto a slide and air-dried. Following this, the slide was stained with Wright-Giemsa composite stain (G1020, Solarbio, China) for 3 min. An equivalent volume of pH 6.4 phosphate buffer was introduced, and the slide was gently agitated to allow mixing with the Wright's stain solution for 5 min. After washing and drying, a microscope (DM4000B, Leica, Germany) was utilized for visualization and image acquisition.

## Western blot

Tissue and cell samples were homogenized on ice after cell lysates and protease inhibitors were added. Centrifugation was performed at 12,000 rpm for 15 min at 4 °C to collect the supernatant. Protein concentration was determined by the BCA detection method. A sample containing 30 μg protein was loaded and run in each well of SDS–PAGE gels. The membranes were incubated with the following antibodies: HO-1 (1:2000, Abcam, ab13243, USA); Ft-L (1:1000, Abcam, ab69090, USA); Ft-H (1:1000, Novus, NBP1-31944, USA); NCOA4 (1:1000, Santa Cruz, sc-373739, USA); TfR (1:1000, Thermo Fisher, 13–6800, USA); Fpn (1:1000, Novus, NBP1-21502, USA); ACSL4 (1:1000, Santa Cruz, sc-271800, USA); xCT (1:1000, Proteintech, 26864–1-AP, USA); Gpx4 (1:1000, Abcam, ab125066, USA); CD206 (1:1000, RD, AF2535, USA); and β-actin (1:1000, Sigma–Aldrich, A5316, USA). The secondary antibodies were as follows: goat anti-mouse (1:10000, Jackson, USA) and goat anti-rabbit (1:10000, Jackson, USA). The gray value of specific blots was scanned and analysed using ImageJ software (National Institute of Health, USA).

## RT–PCR analysis

Total RNA extraction and RT–PCR analysis were performed essentially as described previously (*Luo et al., 2021*). Primer sequences were as follows:

| Gene | Primer sequence (5'–3') | Product length (bp) |
| --- | --- | --- |
| Csf1 | F: GGCTTGGCTTGGGATGATTCT<br>R: GAGGGTCTGGCAGGTACTC | 126 |
| Csf2 | F: GGCCTTGGAAGCATGTAGAGG<br>R: GGAGAACTCGTTAGAGACGACTT | 104 |
| Ccl2 | F: TTAAAAACCTGGATCGGAACCAA<br>R: GCATTAGCTTCAGATTTACGGGT | 121 |
| Ccl7 | F: GCTGCTTTCAGCATCCAAGTG<br>R: CCAGGGACACCGACTACTG | 135 |

## GEO analysis

GSE46480 samples were found in the GEO database of NCBI with the keyword "High Altitude" search. The data set was divided into two groups, one for the plain (Base Camp) and the other for the plateau (Altitude), with a sample size of 98 for each group. Blood samples were collected from the same person at McMurdo Station (48 m) and immediately transferred to Amundsen-Scott South Pole Station (2835 m) on the third day. R language was used for data analysis, and GraphPad Prism was used for statistics.

## Malondialdehyde (MDA), Cysteine (Cys), and Glutathione (GSH) content detection

MDA was detected by a Micro Malondialdehyde Assay Kit (Solarbio, BC0025, China). Cys was detected by a Micro Cysteine Assay Kit (Solarbio, BC0185, China). GSH was detected by a Micro Reduced Glutathione Assay Kit (Solarbio, BC1175, China).

## Immunofluorescence

Spleen sections of 9 µm thickness were flash-frozen and stored at –80 °C. Before immunofluorescence staining, these sections were allowed to acclimate to room temperature for 30 min, then incubated in 10% BSA-PBS for 10 min. Next, the sections were incubated overnight with F4/80-PE (1:200, Biolegend, 123110, USA) or F4/80 (1:100, ab16911, Abcam, USA) and then washed thrice with PBS. Due to the limitations of F4/80 as a definitive macrophage marker, we concurrently conducted immunohistochemical analysis for heme oxygenase-1 (HO-1) (1:200, Abcam, ab13243, USA), CD11b (1:100, 14-0112-82, Thermo Fisher, USA) and CD68 (1:100, ab125212, Abcam, USA), respectively. In the RBCs detection experiment, Band3 primary antibody (1:100, 2813101-AP, Proteintech, China) and PKH67 green fluorescent cell linker (1:250, MIDI67, Sigma, USA) were used. Following this, sections were washed thrice with 0.05% PBST, incubated with Alexa Fluor 555 AffiniPure Donkey Anti-Rabbit IgG (H+L) (1:1000; 711-165-152, Thermo Fisher, USA) at room temperature for 2 hr, washed thrice with PBS again, and finally sealed with 50% glycerine-PBS. Fluorescence microscopy images were captured using a confocal laser scanning microscope (SP8, Leica Microsystems, Wetzlar, Germany). To quantify the immunostaining intensity of F4/80, HO-1, CD11b, CD68, we utilized ImageJ software. Specifically, we captured multiple fields of view for each slide, and the software was used to measure the mean intensity of the fluorescent signal in these images. These measurements were then normalized to the NN group to account for any experimental variations.

## Tissue iron staining (DAB-enhanced Perls' staining)

DAB-enhanced Perls' staining for the spleen paraffin sections was performed as described previously (*Jiang et al., 2022*). Briefly, the sections were washed with PBS and cultured in freshly prepared Perls' solution (1% potassium ferricyanide in 0.1 M hydrochloric acid buffer). The slides were then immersed and stained with DAB. All slides were counterstained with hematoxylin and visualized under a DM4000B microscope (Leica, Germany). Data were collected from three fields of view per mouse and semiquantitatively analysed with ImageJ software. Quantitative results of iron staining were finally normalized to the NN control group.

## Flow cytometry

1. The steps of reticulocyte ratio detection were as follows: Whole blood was extracted from the mice and collected into an anticoagulant tube, which was then set aside for subsequent thiazole orange (TO) staining (*Nébor et al., 2018*). The experimental tube and negative control tube were prepared, 125 µL normal saline, 4 µL anticoagulant and 125 µL TO working solution (1 µg/mL, Sigma, 390062, USA) were added to each tube, and 250 µL normal saline and 4 µL anticoagulant were added to each tube of the negative control tube. After incubation at room temperature for 1 hr, flow cytometry analysis was carried out by using the FL1 (488 nm/525 nm) channel.

2. The steps for the determination of intracellular divalent iron content and lipid peroxidation level were as follows: Splenic tissue was procured from the mice and subsequently processed into a single-cell suspension using a 40 µm filter. The RBCs within the entire sample were subsequently lysed and eliminated, and the remaining cell suspension was resuspended in PBS in preparation for ensuing analyses. A total of $1 \times 10^6$ cells were incubated with 100 µL of BioTracker Far-red Labile $Fe^{2+}$ Dye (1 mM, Sigma, SCT037, USA) for 1 h or C11-BODIPY 581/591 (10 µM, Thermo Fisher, D3861, USA) for 30 min. After the cells were washed with PBS twice, flow cytometry analysis was carried out by using the FL6 (638 nm/660 nm) channel for determination of intracellular divalent iron content or the FL1 (488 nm/525 nm) channel for determination of lipid peroxidation level.

3. The steps for detecting the mortality of spleen cells and macrophages were as follows: $1 \times 10^6$ cells were incubated at room temperature for 40 min with 2 µM Calcein AM and 8 µM Propidium Iodide (PI). Flow cytometry analysis was carried out by using FL1 (488 nm/525 nm, Calcein AM) and FL3 (488 nm/620 nm, PI) channels.

4. The steps for detecting the number of RPMs in the spleen were as follows: $1\times10^6$ cells were incubated with 2% mouse serum-PBS for 10 min, incubated with F4/80-PE (1:1000, 565410, BD, USA) and CD11b PE-CY7 M1/70 (1:1000, 552850, BD, USA) for 30 min, and washed with 2% mouse serum-PBS twice. Flow cytometry analysis was carried out by using FL2 (488 nm/575 nm, F4/80-PE) and FL5 (488 nm/755 nm, CD11b PE-CY7 M1/70) channels.

5. The steps for detecting the number of monocytes in blood, spleen and bone marrow were as follows: $1\times10^6$ cells were incubated with 2% mouse serum-PBS for 10 min, incubated with F4/80-PE, CD11b-PE/CY7 (1:2000, BD, 552850, USA), and Ly6C-APC (1:2000, Thermo Fisher, 17-5932-82) for 30 min, and washed with 2% mouse serum-PBS twice. Flow cytometry analysis was carried out by using FL2 (488 nm/575 nm, F4/80- PE), FL5 (488 nm/755 nm, CD11b-PE/CY7) and FL6 (638 nm/660 nm, Ly6C-APC) channels.

6. The steps employed for assessing erythrophagocytosis in the spleen was scrupulously performed as follows: Initially, RBCs were harvested from mice and subjected to in vitro labelling with PKH67 cell linker. Subsequently, these PKH67-labeled RBCs were administered into mice, followed by exposure to either NN or HH. After a duration of 7 or 14 days of exposure, splenic single-cell suspensions were carefully prepared. From these suspensions, a quantity of $1\times10^6$ cells was isolated and washed in PBS for a period of 5 min. The analytical phase involved flow cytometry, utilizing the FL1 channel (490 nm excitation/502 nm emission) specifically used for detecting PKH67. This approach facilitated the precise quantification and analysis of erythrophagocytosis within the spleen under the specified experimental conditions.

## Phagocytosis of *E. coli* and RBCs

*E. coli* was labelled with Cy5.5 (5 mg/mL), and RBCs were labelled with NHS-biotin (20 mg/mL). Macrophages ($1\times10^6$) were coincubated with *E. coli*-Cy5.5 or RBC-Biotin for 30 min and washed with 2% mouse serum-PBS twice. Flow cytometry analysis was carried out by using FL6 (638 nm/660 nm) to determine the phagocytosis of spleen by *E. coli*. Macrophages (coincubated with Biotin-RBCs) were incubated with streptavidin-FITC for 3 h and washed twice with 2% mouse serum PBS. FL1 (488 nm/525 nm) was used for flow cytometry analysis to determine the phagocytosis of RBCs in the spleen.

## Single-cell RNA sequencing

The spleen tissues were surgically removed and stored in MACS Tissue Storage Solution (Miltenyi Biotec, Bergisch Gladbach, Germany) until processing. Single-cell dissociation was performed by the experimentalists at the GENECHEM laboratory (Shanghai, China). Dissociated single cells were then stained for viability assessment using Calcein-AM (BD Biosciences, USA) and Draq7 (BD Biosciences). The BD Rhapsody system was used to capture transcriptomic information from single cells. Single-cell capture was achieved by random distribution of a single-cell suspension across >200,000 microwells through a limited dilution approach. Beads with oligonucleotide barcodes were added to saturation so that a bead was paired with a cell in a microwell. The cells were lysed in the microwell to hybridize mRNA molecules to barcoded capture oligos on the beads. Beads were collected into a single tube for reverse transcription and ExoI digestion. Upon cDNA synthesis, each cDNA molecule was tagged on the 5′ end (that is, the 3′ end of an mRNA transcript) with a unique molecular identifier (UMI) and cell barcode indicating its cell of origin. Whole transcriptome libraries were prepared using the BD Rhapsody single-cell whole-transcriptome amplification (WTA) workflow, including random priming and extension (RPE), RPE amplification PCR and WTA index PCR. The libraries were quantified using a High Sensitivity D1000 ScreenTape (Agilent) and High Sensitivity D1000 Reagents (Agilent) on a 4150 TapeStation System (Agilent, Palo Alto, CA, USA) and the Qubit High Sensitivity DNA assay (Thermo Fisher Scientific). Sequencing was performed on an Illumina sequencer (Illumina Nova Seq 6000, San Diego, CA) in a 150 bp paired-end run.

## Statistical analysis

Statistical analysis was conducted using GraphPad Prism 8.0 software. All values were represented as the mean ± standard error (SEM). The homogeneity of variance was verified before the application of a parametric test. A two-tailed Student's t-test was employed for data from two groups, while a one-way analysis of variance (ANOVA) with multiple comparisons using Tukey's post hoc test was utilized for data from multiple groups. A p-value of less than 0.05 was considered statistically significant.

## Acknowledgements

Funding This work was supported by Natural Science Foundation of China (Grants 32271228, 81873924 and 82171190), Open Cooperation Program from Key Laboratory of Extreme Environmental Medicine, Ministry of Education (KL2019GY011), Cultivate Candidate of the Jiangsu Province "333" Project.

## Additional information

### Funding

| Funder | Grant reference number | Author |
|---|---|---|
| National Natural Science Foundation of China | 32271228 | Qian-qian Luo |
| National Natural Science Foundation of China | 81873924 | Qian-qian Luo |
| National Natural Science Foundation of China | 82171190 | Guo-hua Wang |
| Key Laboratory of Extreme Environmental Medicine, Ministry of Education | Open Cooperation Program KL2019GY011 | Qian-qian Luo |

The funders had no role in study design, data collection and interpretation, or the decision to submit the work for publication.

### Author contributions

Wan-ping Yang, Data curation, Formal analysis, Funding acquisition; Mei-qi Li, Data curation, Funding acquisition; Jie Ding, Gang Wu, Investigation, Funding acquisition; Jia-yan Li, Formal analysis, Funding acquisition; Bao Liu, Data curation; Yu-qi Gao, Project administration; Guo-hua Wang, Resources, Investigation, Data curation, Project administration, Supervision, Formal analysis, Writing – original draft, Writing – review and editing, Methodology; Qian-qian Luo, Conceptualization, Investigation, Project administration, Supervision, Formal analysis, Visualization, Funding acquisition, Writing – original draft, Writing – review and editing

### Author ORCIDs

Guo-hua Wang https://orcid.org/0000-0002-4810-8534

### Ethics

All animal care and experimental protocols were carried out according to the Chinese Animal Management Rules of the Ministry of Health and were authorized by the Animal Ethics Committees of Nantong University research program protocol #S20190219-011.

Reviewer #2 (Public Review): https://doi.org/10.7554/eLife.87496.4.sa1
Reviewer #3 (Public Review): https://doi.org/10.7554/eLife.87496.4.sa2
Author response https://doi.org/10.7554/eLife.87496.4.sa3

## Additional files

### Supplementary files
• MDAR checklist

### *1234567891Data availability

The scRNA-seq data underpinning the discoveries of this study are archived in the GEO database under the accession code GSE263133. The publicly available reused data comes from the GEO database at NCBI, specifically the GSE46480 samples, which were found through a search using the keyword "High Altitude". Other data generated or analysed during this study are included in the manuscript and supporting files.

The following dataset was generated:

| Author(s) | Year | Dataset title | Dataset URL | Database and Identifier |
|---|---|---|---|---|
| Yang W, Li M, Ding J, Li J, Wang G, Luo Q | 2024 | High-altitude hypoxia exposure inhibits erythrophagocytosis by inducing macrophage ferroptosis in the spleen | https://www.ncbi.nlm.nih.gov/geo/query/acc.cgi?acc=GSE263133 | NCBI Gene Expression Omnibus, GSE263133 |

The following previously published dataset was used:

| Author(s) | Year | Dataset title | Dataset URL | Database and Identifier |
|---|---|---|---|---|
| Herman NM, Grill DE, Anderson PJ, Miller AD, Johnson JB, O'Malley KA, Ceridon Richert ML, Johnson BD | 2013 | Peripheral blood mononuclear cell (PBMC) gene expression in healthy adults rapidly transported to high altitude | https://www.ncbi.nlm.nih.gov/geo/query/acc.cgi?acc=GSE46480 | NCBI Gene Expression Omnibus, GSE46480 |

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
