## [Editor Report · eLife assessment]

This **useful** study reports that a week or more of hypoxia exposure in mice increases erythropoiesis and decreases the number of iron-recycling macrophages in the spleen, compromising their capacity for red blood cell phagocytosis – reflected by increased mature erythrocyte retention in the spleen. Compared to an earlier version, the study has been strengthened with mouse experiments under hypobaric hypoxia and complemented by extensive ex vivo analyses. Unfortunately, while some of the evidence is **solid**, the work as it currently stands only **incompletely** supports the authors' hypotheses. While the study would benefit from additional experiments that more directly buttress the central claims, it should be of interest to the fields of hemopoiesis and bone marrow biology and possibly also blood cancer.

---

## [Referee Report · Reviewer #2 (Public Review)]

The authors aimed at elucidating the development of high altitude polycythemia which affects mice and men staying in a hypoxic atmosphere at high altitude (hypobaric hypoxia; HH). HH causes increased erythropoietin production which stimulates the production of red blood cells. The authors hypothesize that increased production is only partially responsible for exaggerated red blood cell production, i.e. polycythemia, but that decreased erythrophagocytosis in the spleen contributes to high red blood cells counts.

The main strength of the study is the use of a mouse model exposed to HH in a hypobaric chamber. However, not all of the reported results are convincing due to some smaller effects which one may doubt to result in the overall increase in red blood cells as claimed by the authors. Moreover, direct proof for reduced erythrophagocytosis is compromised due to a strong spontaneous loss of labelled red blood cells, although effects of labelled *E. coli* phagocytosis are shown.

Comments on latest version:

The authors have partly addressed my comments.

(1) The response to my question regarding unchanged MCH is a kind of "hand waiving" - maybe it would require substantially more extensive work to clarify this issue

(2) The moderate if not marginal difference in normal vs splenectomy argues against a significant role of the spleen - even if the difference was slightly larger in HH

(3) There is still overinterpretation of data. My Q was: Is the reduced phagocytic capacity in Fig 4B significant? Response: "This is indicative of a diminished phagocytic capacity, particularly when contrasted

with NN conditions." I guess that is a "no"

(4) I assume my question with respect to bi- or trivalent iron chelators was misunderstood.

In general, as indicated above, it is an interesting hypothesis which is corroborated by data in several instances. Maybe the scientific community should decide whether it is all in all conclusive.

---

## [Referee Report · Reviewer #3 (Public Review)]

The manuscript by Yang et al. investigated in mice how hypobaric hypoxia can modify the RBC clearance function of the spleen, a concept that is of interest. Via interpretation of their data, the authors proposed a model that hypoxia causes an increase in cellular iron levels, possibly in RPMs, leading to ferroptosis, and downregulates their erythrophagocytic capacity.

Comments on revised version:

The manuscript has now improved with all the new data, supporting the model proposed by the authors. However, it remains not very easy to follow for the conclusions and experimental details. Some of the most important remaining comments are listed below:

(1) Lines 401-406 - The conclusions in this new fragment sound a bit overstated - the authors do not directly measure erytrophagocytosis capacity, only the total RBC parameters in the circulation. The increase is also very mild biologically between sham and splenectomized mice in HH conditions.

(2) scRNA seq data are still presented in a way that is very difficult to understand. The readers could not see from the graphics that macrophages are depleted. The clusters are not labelled - some clusters in the bin 'macrophahes+DC' seem actually to be more represented in Fig. 3E; Fig. 3F does not correspond to Fig. 3D. It would be maybe more informative to present like in Figure D side by side NN versus HH? The authors could consider moving the data from supplements that relate to RPMs to the main figure and making it consistent for the Clusters - eg, the authors show data for Cluster 0 in the supplement, and the same Cluster is not marked as macrophages in the main figure. This is quite difficult to follow.

(3) Figure 3G has likely mislabeled axis for F4/80 and CD11b - such mistakes should be avoided in a second revised version of the manuscript, and this data is now redundant with the data shown as new Figure 5A.

(4) The data from new Figure 4 should be better mentioned in the main body of the manuscript - all panels are mentioned twice in the text, first speaking about the decline of labelled RBCs and second referring to phagocytic capacity, whereas this figure only illustrates the decline of labelled RBCs, not directly phagocytic capacity of RPMs. What is lacking, as opposed to typical RBC life span assay, is the time '0' ('starting point') - this is particularly important as we can observe a big drop in labelled RBCs for eg 7 days between NN and HH group, actually implying increased removal of labelled RBCs within the first days of hypoxia exposure. What should be better labelled in this figure is that the proportion of RBCs are labelled RBCs not all RBCs (Y axis in individual panels). Overall, the new Figure 4 brings new data to the study, but how it is presented and discussed is not at the 'state-of-the-art' level (eg, missing the time '0') and is not very straightforward to the reader.

(5) In Figure 7, the experiments with Tuftsin are not very easy to follow, especially for the major conclusions. In panels A and B, the focus is the drug itself under NN conditions, with RBC removal as a readout. Then, in the next panels, the authors introduce HH, and then look at the F4/80 and iron staining. What was exactly the major point the authors wanted to make here?

(6) The data from Figure 8 are informative but do not address the individual cell types - eg, a drop in HO1 or FT may be due to the depletion of RPMs. An increase of TFR1 could be due to the retention of RBCs, the same as maybe labile iron. The data from PBMC are only very loosely linked to these phenotypes observed in the total spleen, and the reason for the regulation of the same proteins in PBMC might be different. It goes back to the data in Figure 3A-C, where also total splenocytes are investigated for their viability.

(7) Can the authors provide the data for the purity (eg cell surface markers) of their primary splenic macrophage cultures? Only ensuring that these are macrophages or addressing the readouts from Figure 8 in RPMs could link ferroptosis to RPMs under HH conditions.

(8) All the data are not presented as individual data points which is not widely applied in papers.

(9) No gating strategies are nicely illustrated or described.

---

## [Author Response]

The following is the authors’ response to the previous reviews.

**Recommendations for the authors:**
(1) Substantial revision of the claims and interpretation of the results is needed, especially in the setting of additional data showing enhanced erythrophagocytosis with decreased RBC lifespan.

Thank you for your valuable feedback and suggestion for a substantial revision of the claims and interpretation of our results. We acknowledge the importance of considering additional data that shows enhanced erythrophagocytosis with decreased RBC lifespan. In response, we have revised our manuscript and incorporated additional experimental data to support and clarify our findings.

(1) In our original manuscript, we reported a decrease in the number of splenic red pulp macrophages (RPMs) and phagocytic erythrocytes after hypobaric hypoxia (HH) exposure. This conclusion was primarily based on our observations of reduced phagocytosis in the spleen.

(2) Additional experimental data on RBC labeling and erythrophagocytosis:

Experiment 1 (RBC labeling and HH exposure)

We conducted an experiment where RBCs from mice were labeled with PKH67 and injected back into the mice. These mice were then exposed to normal normoxia (NN) or HH for 7 or 14 days. The subsequent assessment of RPMs in the spleen using flow cytometry and immunofluorescence detection revealed a significant decrease in both the population of splenic RPMs (F4/80hiCD11blo, new Figure 5A and C) and PKH67-positive macrophages after HH exposure (as depicted in new Figure 5A and C-E). This finding supports our original claim of reduced phagocytosis under HH conditions.

-Experiment 2 (erythrophagocytosis enhancement)

To examine the effects of enhanced erythrophagocytosis, we injected Tuftsin after administering PKH67-labelled RBCs. Our observations showed a significant decrease in PKH67 fluorescence in the spleen, particularly after Tuftsin injection compared to the NN group. This result suggests a reduction in RBC lifespan when erythrophagocytosis is enhanced (illustrated in new Figure 7, A-B).

**Author response image 2. sa3fig2:** 

(3) Revised conclusions:

The additional data from these experiments support our original findings by providing a more comprehensive view of the impact of HH exposure on splenic erythrophagocytosis.The decrease in phagocytic RPMs and phagocytic erythrocytes after HH exposure, along with the observed decrease in RBC lifespan following enhanced erythrophagocytosis, collectively suggest a more complex interplay between hypoxia, erythrophagocytosis, and RBC lifespan than initially interpreted.

We think that these revisions and additional experimental data provide a more robust and detailed understanding of the effects of HH on splenic erythrophagocytosis and RBCs lifespan. We hope that these changes adequately address the concerns raised and strengthen the conclusions drawn in our manuscript.

(2) F4/80 high; CD11b low are true RPMs which the cells which the authors are presenting, i.e. splenic monocytes / pre-RPMs. To discuss RPM function requires the presentation of these cells specifically rather than general cells in the proper area of the spleen.

Thank you for your feedback requesting a substantial revision of our claims and interpretation, particularly considering additional data showing enhanced erythrophagocytosis with decreased RBC lifespan. In response, we have thoroughly revised our manuscript and included new experimental data that further elucidate the effects of HH on RPMs and erythrophagocytosis.

(1) Re-evaluation of RPMs population after HH exposure:

Flow cytometry analysis (new Figure 3G, Figure 5A and B): We revisited the analysis of RPMs (F4/80hiCD11blo) in the spleen after 7 and 14 days of HH exposure. Our revised flow cytometry data consistently showed a significant decrease in the RPMs population post-HH exposure, reinforcing our initial findings.

**Author response image 3. sa3fig3:** 

**Author response image 4. sa3fig4:** 

In situ expression of RPMs (Figure S1, A-D):

We further confirmed the decreased population of RPMs through in situ co-staining with F4/80 and CD11b, and F4/80 and CD68, in spleen tissues. These results clearly demonstrated a significant reduction in F4/80hiCD11blo (Figure S1, A and B) and F4/80hiCD68hi (Figure S1, C and D) cells following HH exposure.

**Author response image 5. sa3fig5:** 

(2) Single-cell sequencing analysis of splenic RPMs:

We conducted a single-cell sequencing analysis of spleen samples post 7 days of HH exposure (Figure S2, A-C). This analysis revealed a notable shift in the distribution of RPMs, predominantly associated with Cluster 0 under NN conditions, to a reduced presence in this cluster after HH exposure.Pseudo-time series analysis indicated a transition pattern change in spleen RPMs, with a shift from Cluster 2 and Cluster 1 towards Cluster 0 under NN conditions, and a reverse transition following HH exposure (Figure S2, B and D). This finding implies a decrease in resident RPMs in the spleen under HH conditions.

(3) Consolidated findings and revised interpretation:

The comprehensive analysis of flow cytometry, in situ staining, and single-cell sequencing data consistently indicates a significant reduction in the number of RPMs following HH exposure.These findings, taken together, strongly support the revised conclusion that HH exposure leads to a decrease in RPMs in the spleen, which in turn may affect erythrophagocytosis and RBC lifespan.

**Author response image 6. sa3fig6:** 

In conclusion, our revised manuscript now includes additional experimental data and analyses, strengthening our claims and providing a more nuanced interpretation of the impact of HH on spleen RPMs and related erythrophagocytosis processes. We believe these revisions and additional data address your concerns and enhance the scientific validity of our study.

(3) RBC retention in the spleen should be measured anyway quantitatively, eg, with proper flow cytometry, to determine whether it is increased or decreased.

Thank you for your query regarding the quantitative measurement of RBC retention in the spleen, particularly in relation to HH exposure. We have utilized a combination of techniques, including flow cytometry and histological staining, to investigate this aspect comprehensively. Below is a summary of our findings and methodology.

(1) Flow cytometry analysis of labeled RBCs:

Our study employed both NHS-biotin (new Figure 4, A-D) and PKH67 labeling (new Figure 4, E-H) to track RBCs in mice exposed to HH. Flow cytometry results from these experiments (new Figure 4, A-H) showed a decrease in the proportion of labeled RBCs over time, both in the blood and spleen. Notably, there was a significantly greater reduction in the amplitude of fluorescently labeled RBCs after NN exposure compared to the reduced amplitude of fluorescently labeled RBCs observed in blood and spleen under HH exposure. The observed decrease in labeled RBCs was initially counterintuitive, as we expected an increase in RBC retention due to reduced erythrophagocytosis. However, this decrease can be attributed to the significantly increased production of RBCs following HH exposure, diluting the proportion of labeled cells.Specifically, for blood, the biotin-labeled RBCs decreased by 12.06% under NN exposure and by 7.82% under HH exposure, while the PKH67-labeled RBCs decreased by 9.70% under NN exposure and by 4.09% under HH exposure. For spleen, the biotin-labeled RBCs decreased by 3.13% under NN exposure and by 0.46% under HH exposure, while the PKH67-labeled RBCs decreased by 1.16% under NN exposure and by 0.92% under HH exposure. These findings suggest that HH exposure leads to a decrease in the clearance rate of RBCs.

**Author response image 7. sa3fig7:** 

(2) Detection of erythrophagocytosis in spleen:

To assess erythrophagocytosis directly, we labeled RBCs with PKH67 and analyzed their uptake by splenic macrophages (F4/80hi) after HH exposure. Our findings (new Figure 5, D-E) indicated a decrease in PKH67-positive macrophages in the spleen, suggesting reduced erythrophagocytosis.

**Author response image 8. sa3fig8:** 

(3) Flow cytometry analysis of RBC retention:

Our flow cytometry analysis revealed a decrease in PKH67-positive RBCs in both blood and spleen (Figure S4). We postulated that this was due to increased RBC production after HH exposure. However, this method might not accurately reflect RBC retention, as it measures the proportion of PKH67-labeled RBCs relative to the total number of RBCs, which increased after HH exposure.

**Author response image 9. sa3fig9:** 

(4) Histological and immunostaining analysis:

Histological examination using HE staining and band3 immunostaining in situ (new Figure 6, A-D, and G-H) revealed a significant increase in RBC numbers in the spleen after HH exposure. This was further confirmed by detecting retained RBCs in splenic single cells using Wright-Giemsa composite stain (new Figure 6, E and F) and retained PKH67-labelled RBCs in spleen (new Figure 6, I and J).

**Author response image 10. sa3fig10:** 

(5) Interpreting the data:

The comprehensive analysis suggests a complex interplay between increased RBC production and decreased erythrophagocytosis in the spleen following HH exposure. While flow cytometry indicated a decrease in the proportion of labeled RBCs, histological and immunostaining analyses demonstrated an actual increase in RBCs retention in the spleen. These findings collectively suggest that while the overall RBCs production is upregulated following HH exposure, the spleen's capacity for erythrophagocytosis is concurrently diminished, leading to increased RBCs retention.

(6) Conclusion:

Taken together, our results indicate a significant increase in RBCs retention in the spleen post-HH exposure, likely due to reduced residual RPMs and erythrophagocytosis. This conclusion is supported by a combination of flow cytometry, histological staining, and immunostaining techniques, providing a comprehensive view of RBC dynamics under HH conditions. We think these findings offer a clear quantitative measure of RBC retention in the spleen, addressing the concerns raised in your question.

(4) Numerous other methodological problems as listed below.

We appreciate your question, which highlights the importance of using multiple analytical approaches to understand complex physiological processes. Please find below our point-by-point response to the methodological comments.

**Reviewer #1 (Recommendations For The Authors):**
(1) Decreased BM and spleen monocytes d/t increased liver monocyte migration is unclear. there is no evidence that this happens or why it would be a reasonable hypothesis, even in splenectomized mice.

Thank you for highlighting the need for further clarification and justification of our hypothesized decrease in BM and spleen monocytes due to increased monocyte migration to the liver, particularly in the context of splenectomized mice. Indeed, our study has not explicitly verified an augmentation in mononuclear cell migration to the liver in splenectomized mice.

Nonetheless, our investigations have revealed a notable increase in monocyte migration to the liver after HH exposure. Noteworthy is our discovery of a significant upregulation in colony stimulating factor-1 (CSF-1) expression in the liver, observed after both 7 and 14 days of HH exposure (data not included). This observation was substantiated through flow cytometry analysis (as depicted in Figure S4), which affirmed an enhanced migration of monocytes to the liver. Specifically, we noted a considerable increase in the population of transient macrophages, monocytes, and Kupffer cells in the liver following HH exposure.

**Author response image 11. sa3fig11:** 

Considering these findings, we hypothesize that hypoxic conditions may activate a compensatory mechanism that directs monocytes towards the liver, potentially linked to the liver’s integral role in the systemic immune response. In accordance with these insights, we intend to revise our manuscript to reflect the speculative nature of this hypothesis more accurately, and to delineate the strategies we propose for its further empirical investigation. This amendment ensures that our hypothesis is presented with full consideration of its speculative basis, supported by a coherent framework for future validation.

(2) While F4/80+CD11b+ population is decreased, this is mainly driven by CD11b and F4/80+ alone population is significantly increased. This is counter to the hypothesis.

Thank you for addressing the apparent discrepancy in our findings concerning the F4/80+CD11b+ population and the increase in the F4/80+ alone population, which seems to contradict our initial hypothesis. Your observation is indeed crucial for the integrity of our study, and we appreciate the opportunity to clarify this matter.

(1) Clarification of flow cytometry results:

In response to the concerns raised, we revisited our flow cytometry experiments with a focus on more clearly distinguishing the cell populations. Our initial graph had some ambiguities in cell grouping, which might have led to misinterpretations.The revised flow cytometry analysis, specifically aimed at identifying red pulp macrophages (RPMs) characterized as F4/80hiCD11blo in the spleen, demonstrated a significant decrease in the F4/80 population. This finding is now in alignment with our immunofluorescence results.

**Author response image 12. sa3fig12:** 

**Author response image 13. sa3fig13:** 

(2) Revised data and interpretation:

The results presented in new Figure 3G and Figure 5 (A and B) consistently indicate a notable reduction in the RPMs population following HH exposure. This supports our revised understanding that HH exposure leads to a decrease in the specific macrophage subset (F4/80hiCD11blo) in the spleen.

We’ve updated our manuscript to reflect these new findings and interpretations. The revised manuscript details the revised flow cytometry analysis and discusses the potential mechanisms behind the observed changes in macrophage populations.

(3) HO-1 expression cannot be used as a surrogate to quantify number of macrophages as the expression per cell can decrease and give the same results. In addition, the localization of effect to the red pulp is not equivalent to an assertion that the conclusion applies to macrophages given the heterogeneity of this part of the organ and the spleen in general.

Thank you for your insightful comments regarding the use of HO-1 expression as a surrogate marker for quantifying macrophage numbers, and for pointing out the complexity of attributing changes in HO-1 expression specifically to macrophages in the splenic red pulp. Your observations are indeed valid and warrant a detailed response.

(1) Role of HO-1 in macrophage activity:

In our study, HO-1 expression was not utilized as a direct marker for quantifying macrophages. Instead, it was considered an indicator of macrophage activity, particularly in relation to erythrophagocytosis. HO-1, being upregulated in response to erythrophagocytosis, serves as an indirect marker of this process within splenic macrophages.The rationale behind this approach was that increased HO-1 expression, induced by erythrophagocytosis in the spleen’s red pulp, could suggest an augmentation in the activity of splenic macrophages involved in this process.

(2) Limitations of using HO-1 as an indicator:

We acknowledge your point that HO-1 expression per cell might decrease, potentially leading to misleading interpretations if used as a direct quantifier of macrophage numbers. The variability in HO-1 expression per cell indeed presents a limitation in using it as a sole indicator of macrophage quantity.Furthermore, your observation about the heterogeneity of the spleen, particularly the red pulp, is crucial. The red pulp is a complex environment with various cell types, and asserting that changes in HO-1 expression are exclusive to macrophages could oversimplify this complexity.

(3) Addressing the concerns:

To address these concerns, we propose to supplement our HO-1 expression data with additional specific markers for macrophages. This would help in correlating HO-1 expression more accurately with macrophage numbers and activity.We also plan to conduct further studies to delineate the specific cell types in the red pulp contributing to HO-1 expression. This could involve techniques such as immunofluorescence or immunohistochemistry, which would allow us to localize HO-1 expression to specific cell populations within the splenic red pulp.

We’ve revised our manuscript to clarify the role of HO-1 expression as an indirect marker of erythrophagocytosis and to acknowledge its limitations as a surrogate for quantifying macrophage numbers.

(4) line 63-65 is inaccurate as red cell homeostasis reaches a new steady state in chronic hypoxia.

Thank you for pointing out the inaccuracy in lines 63-65 of our manuscript regarding red cell homeostasis in chronic hypoxia. Your feedback is invaluable in ensuring the accuracy and scientific integrity of our work. We’ve revised lines 63-65 to accurately reflect the understanding.

(5) Eryptosis is not defined in the manuscript.

Thank you for highlighting the omission of a definition for eryptosis in our manuscript. We acknowledge the significance of precisely defining such key terminologies, particularly when they play a crucial role in the context of our research findings.Eryptosis, a term referenced in our study, is a specialized form of programmed cell death unique to erythrocytes. Similar with apoptosis in other cell types, eryptosis is characterized by distinct physiological changes including cell shrinkage, membrane blebbing, and the externalization of phosphatidylserine on the erythrocyte surface. These features are indicative of the RBCs lifecycle and its regulated destruction process.

However, it is pertinent to note that our current study does not extensively delve into the mechanisms or implications of eryptosis. Our primary focus has been to elucidate the effects of HH exposure on the processes of splenic erythrophagocytosis and the resultant impact on the lifespan of RBCs. Given this focus, and to maintain the coherence and relevance of our manuscript, we have decided to exclude specific discussions of eryptosis from our revised manuscript. This decision aligns with our aim to provide a clear and concentrated exploration of the influence of HH exposure on RBCs dynamics and splenic function.

We appreciate your input, which has significantly contributed to enhancing the clarity and accuracy of our manuscript. The revision ensures that our research is presented with a focused scope, aligning closely with our experimental investigations and findings.

(6) Physiologically, there is no evidence that there is any "free iron" in cells, making line 89 point inaccurate.

Thank you for highlighting the concern regarding the reference to "free iron" in cells in line 89 of our manuscript. The term "free iron" in our manuscript was intended to refer to divalent iron (Fe2+), rather than unbound iron ions freely circulating within cells. We acknowledge that the term "free iron" might lead to misconceptions, as it implies the presence of unchelated iron, which is not physiologically common due to the potential for oxidative damage. To rectify this and provide clarity, we’ve revised line 89 of our manuscript to reflect our meaning more accurately. Instead of "free iron," we use "divalent iron (Fe2+)" to avoid any misunderstanding regarding the state of iron in cells.We also ensure that any implications drawn from the presence of Fe2+ in cells are consistent with current scientific literature and understanding.

(7) Fig 1f no stats

We appreciate your critical review and suggestions, which help in improving the accuracy and clarity of our research. We’ve revised statistic diagram of new Figure 1F.

(8) Splenectomy experiments demonstrate that erythrophagocytosis is almost completely replaced by functional macrophages in other tissues (likely Kupffer cells in the liver). there is only a minor defect and no data on whether it is in fact the liver or other organs that provide this replacement function and makes the assertions in lines 345-349 significantly overstated.

Thank you for your critical assessment of our interpretation of the splenectomy experiments, especially concerning the role of erythrophagocytosis by macrophages in other tissues, such as Kupffer cells in the liver. We appreciate your observation that our assertions may be overstated and acknowledge the need for more specific data to identify which organs compensate for the loss of splenic erythrophagocytosis.

(1) Splenectomy experiment findings:

Our findings in Figure 2D do indicate that in the splenectomized group under NN conditions, erythrophagocytosis is substantially compensated for by functional macrophages in other tissues. This is an important observation that highlights the body's ability to adapt to the loss of splenic function.However, under HH conditions, our data suggest that the spleen plays an important role in managing erythrocyte turnover, as indicated by the significant impact of splenectomy on erythrophagocytosis and subsequent erythrocyte dynamics.

(2) Addressing the lack of specific organ identification:

We acknowledge that our study does not definitively identify which organs, such as the liver or others, take over the erythrophagocytosis function post-splenectomy. This is an important aspect that needs further investigation.To address this, we also plan to perform additional experiments that could more accurately point out the specific tissues compensating for the loss of splenic erythrophagocytosis. This could involve tracking labeled erythrocytes or using specific markers to identify macrophages actively engaged in erythrophagocytosis in various organs.

(3) Revising manuscript statements:

Considering your feedback, we’ve revised the statements in lines 345-349 (lines 378-383 in revised manuscript) to enhance the scientific rigor and clarity of our research presentation.

(9) M1 vs M2 macrophage experiments are irrelevant to the main thrust of the manuscript, there are no references to support the use of only CD16 and CD86 for these purposes, and no stats are provided. It is also unclear why bone marrow monocyte data is presented and how it is relevant to the rest of the manuscript.

Thank you for your critical evaluation of the relevance and presentation of the M1 vs. M2 macrophage experiments in our manuscript. We appreciate your insights, especially regarding the use of specific markers and the lack of statistical analysis, as well as the relevance of bone marrow monocyte data to our study's main focus.

(1) Removal of M1 and M2 macrophage data:

Based on your feedback and our reassessment, we agree that the results pertaining to M1 and M2 macrophages did not align well with the main objectives of our manuscript. Consequently, we have decided to remove the related content on M1 and M2 macrophages from the revised manuscript. This decision was made to ensure that our manuscript remains focused and coherent, highlighting our primary findings without the distraction of unrelated or insufficiently supported data.

The use of only CD16 and CD86 markers for M1 and M2 macrophage characterization, without appropriate statistical analysis, was indeed a methodological limitation. We recognize that a more comprehensive set of markers and rigorous statistical analysis would be necessary for a meaningful interpretation of M1/M2 macrophage polarization. Furthermore, the relevance of these experiments to the central theme of our manuscript was not adequately established. Our study primarily focuses on erythrophagocytosis and red pulp macrophage dynamics under hypobaric hypoxia, and the M1/M2 polarization aspect did not contribute significantly to this narrative.

(2) Clarification on bone marrow monocyte data:

Regarding the inclusion of bone marrow monocyte data, we acknowledge that its relevance to the main thrust of the manuscript was not clearly articulated. In the revised manuscript, we provide a clearer rationale for its inclusion and how it relates to our primary objectives.

(3) Commitment to clarity and relevance:

We are committed to ensuring that every component of our manuscript contributes meaningfully to our overall objectives and research questions. Your feedback has been instrumental in guiding us to streamline our focus and present our findings more effectively.

We appreciate your valuable feedback, which has led to a more focused and relevant presentation of our research. These changes enhance the clarity and impact of our manuscript, ensuring that it accurately reflects our key research findings.

(10) Biotinolated RBC clearance is enhanced, demonstrating that RBC erythrophagocytosis is in fact ENHANCED, not diminished, calling into question the founding hypothesis that the manuscript proposes.

Thank you for your critical evaluation of our data on biotinylated RBC clearance, which suggests enhanced erythrophagocytosis under HH conditions. This observation indeed challenges our founding hypothesis that erythrophagocytosis is diminished in this setting. Below is a summary of our findings and methodology.

(1) Interpretation of RBC labeling results:

Both the previous results of NHS-biotin labeled RBCs (new Figure 4, A-D) and the current results of PKH67-labeled RBCs (new Figure 4, E-H) demonstrated a decrease in the number of labeled RBCs with an increase in injection time. The production of RBCs, including bone marrow and spleen production, was significantly increased following HH exposure, resulting in a consistent decrease in the proportion of labeled RBCs via flow cytometry detection both in the blood and spleen of mice compared to the NN group. However, compared to the reduced amplitude of fluorescently labeled RBCs observed in blood and spleen under NN exposure, there was a significantly weaker reduction in the amplitude of fluorescently labeled RBCs after HH exposure. Specifically, for blood, the biotin-labeled RBCs decreased by 12.06% under NN exposure and by 7.82% under HH exposure, while the PKH67-labeled RBCs decreased by 9.70% under NN exposure and by 4.09% under HH exposure. For spleen, the biotin-labeled RBCs decreased by 3.13% under NN exposure and by 0.46% under HH exposure, while the PKH67-labeled RBCs decreased by 1.16% under NN exposure and by 0.92% under HH exposure.

**Author response image 14. sa3fig14:** 

(2) Increased RBCs production under HH conditions:

It's important to note that RBCs production, including from bone marrow and spleen, was significantly increased following HH exposure. This increase in RBCs production could contribute to the decreased proportion of labeled RBCs observed in flow cytometry analyses, as there are more unlabeled RBCs diluting the proportion of labeled cells in the blood and spleen.

(3) Analysis of erythrophagocytosis in RPMs:

Our analysis of PKH67-labeled RBCs content within RPMs following HH exposure showed a significant reduction in the number of PKH67-positive RPMs in the spleen (new Figure 5). This finding suggests a decrease in erythrophagocytosis by RPMs under HH conditions.

**Author response image 15. sa3fig15:** 

(4) Reconciling the findings:

The apparent contradiction between enhanced RBC clearance (suggested by the reduced proportion of labeled RBCs) and reduced erythrophagocytosis in RPMs (indicated by fewer PKH67-positive RPMs) may be explained by the increased overall production of RBCs under HH. This increased production could mask the actual erythrophagocytosis activity in terms of the proportion of labeled cells. Therefore, while the proportion of labeled RBCs decreases more significantly under HH conditions, this does not necessarily indicate an enhanced erythrophagocytosis rate, but rather an increased dilution effect due to higher RBCs turnover.

(5) Revised interpretation and manuscript changes:

Given these factors, we update our manuscript to reflect this detailed interpretation and clarify the implications of the increased RBCs production under HH conditions on our observations of labeled RBCs clearance and erythrophagocytosis. We appreciate your insightful feedback, which has prompted a careful re-examination of our data and interpretations. We hope that these revisions provide a more accurate and comprehensive understanding of the effects of HH on erythrophagocytosis and RBCs dynamics.

(11) Legend in Fig 4c-4d looks incorrect and Fig 4e-4f is very non-specific since Wright stain does not provide evidence of what type of cells these are and making for a significant overstatement in the contribution of this data to "confirming" increased erythrophagocytosis in the spleen under HH exposure (line 395-396).

Thank you for your insightful observations regarding the data presentation and figure legends in our manuscript, particularly in relation to Figure 4 (renamed as Figure 6 in the revised manuscript) and the use of Wright-Giemsa composite staining. We appreciate your constructive feedback and acknowledge the importance of presenting our data with utmost clarity and precision.

(1) Amendments to Figure legends:

We recognize the necessity of rectifying inaccuracies in the legends of the previously labeled Figure 4C and D. Corrections have been meticulously implemented to ensure the legends accurately contain the data presented. Additionally, we acknowledge the error concerning the description of Wright staining. The method employed in our study is Wright-Giemsa composite staining, which, unlike Wright staining that solely stains cytoplasm (RBC), is capable of staining both nuclei and cytoplasm.

(2) Addressing the specificity of Wright-Giemsa Composite staining:

Our approach involved quantifying RBC retention using Wright-Giemsa composite staining on single splenic cells post-perfusion at 7 and 14 days post HH exposure. We understand and appreciate your concerns regarding the nonspecific nature of Wright staining. Although Wright stain is a general hematologic stain and not explicitly specific for certain cell types, its application in our study aimed to provide preliminary insights. The spleen cells, devoid of nuclei and thus likely to be RBCs, were stained and observed post-perfusion, indicating RBC retention within the spleen.

(3) Incorporating additional methods for RBC identification:

To enhance the specificity of our findings, we integrated supplementary methods for RBC identification in the revised manuscript. We employed band3 immunostaining (in the new Figure 6, C-D and G-H) and PKH67 labeling (Figure 6, I-J) for a more targeted identification of RBCs. Band3, serving as a reliable marker for RBCs, augments the specificity of our immunostaining approach. Likewise, PKH67 labeling affords a direct and definitive means to assess RBC retention in the spleen following HH exposure.

**Author response image 16. sa3fig16:** 

(4) Revised interpretation and manuscript modifications:

Based on these enhanced methodologies, we have refined our interpretation of the data and accordingly updated the manuscript. The revised narrative underscores that our conclusions regarding reduced erythrophagocytosis and RBC retention under HH conditions are corroborated by not only Wright-Giemsa composite staining but also by band3 immunostaining and PKH67 labeling, each contributing distinctively to our comprehensive understanding.

We are committed to ensuring that our manuscript precisely reflects the contribution of each method to our findings and conclusions. Your thorough review has been invaluable in identifying and rectifying areas for improvement in our research report and interpretation.

(12) Ferroptosis data in Fig 5 is not specific to macrophages and Fer-1 data confirms the expected effect of Fer-1 but there is no data that supports that Fer-1 reverses the destruction of these cells or restores their function in hypoxia. Finally, these experiments were performed in peritoneal macrophages which are functionally distinct from splenic RPM.

Thank you for your critique of our presentation and interpretation of the ferroptosis data in Figure 5 (renamed as Figure 9 in the revised manuscript), as well as your observations regarding the specificity of the experiments to macrophages and the effects of Fer-1. We value your input and acknowledge the need to clarify these aspects in our manuscript.

(1) Clarification on cell type used in experiments:

We appreciate your attention to the details of our experimental setup. The experiments presented in Figure 9 were indeed conducted on splenic macrophages, not peritoneal macrophages, as incorrectly mentioned in the original figure legend. This was an error in our manuscript, and we have revised the figure legend accordingly to accurately reflect the cell type used.

(2) Specificity of ferroptosis data:

We recognize that the data presented in Figure 9 need to be more explicitly linked to the specific macrophage population being studied. In the revised manuscript, we ensure that the discussion around ferroptosis data is clearly situated within the framework of splenic macrophages.We also provide additional methodological details in the 'Methods' section to reinforce the specificity of our experiments to splenic macrophages.

(3) Effects of Fer-1 on macrophage function and survival:

Regarding the effect of Fer-1, we agree that while our data confirms the expected effect of Fer-1 in inhibiting ferroptosis, we have not provided direct evidence that Fer-1 reverses the destruction of macrophages or restores their function in hypoxia.To address this, we propose additional experiments to specifically investigate the impact of Fer-1 on the survival and functional restoration of splenic macrophages under hypoxic conditions. This would involve assessing not only the inhibition of ferroptosis but also the recovery of macrophage functionality post-treatment.

(4) Revised interpretation and manuscript changes:

We’ve revised the relevant sections of our manuscript to reflect these clarifications and proposed additional studies. This includes modifying the discussion of the ferroptosis data to more accurately represent the cell types involved and the limitations of our current findings regarding the effects of Fer-1.The revised manuscript presents a more detailed interpretation of the ferroptosis data, clearly describing what our current experiments demonstrate and what remains to be investigated.

We are grateful for your insightful feedback, which has highlighted important areas for improvement in our research presentation. We think that these revisions will enhance the clarity and scientific accuracy of our manuscript, ensuring that our findings and conclusions are well-supported and precisely communicated.

**Reviewer #2 (Recommendations For The Authors):**
The following questions and remarks should be considered by the authors:(1) The methods should clearly state whether the HH was discontinued during the 7 or 14 day exposure for cleaning, fresh water etc. Moreover, how was CO2 controlled? The procedure for splenectomy needs to be described in the methods.

Thank you for your inquiry regarding the specifics of our experimental methods, particularly the management of HH exposure and the procedure for splenectomy. We appreciate your attention to detail and the importance of these aspects for the reproducibility and clarity of our research.

(1) HH exposure conditions:

In our experiments, mice were continuously exposed to HH for the entire duration of 7 or 14 days, without interruption for activities such as cleaning or providing fresh water. This uninterrupted exposure was crucial for maintaining consistent hypobaric conditions throughout the experiment. The hypobaric chamber was configured to ensure a ventilation rate of 25 air exchanges per minute. This high ventilation rate was effective in regulating the concentration of CO2 inside the chamber, thereby maintaining a stable environment for the mice.

(2) The splenectomy was performed as follows:

After anesthesia, the mice were placed in a supine position, and their limbs were fixed. The abdominal operation area was skinned, disinfected, and covered with a sterile towel. A median incision was made in the upper abdomen, followed by laparotomy to locate the spleen. The spleen was then carefully pulled out through the incision. The arterial and venous directions in the splenic pedicle were examined, and two vascular forceps were used to clamp all the tissue in the main cadre of blood vessels below the splenic portal. The splenic pedicle was cut between the forceps to remove the spleen. The end of the proximal hepatic artery was clamped with a vascular clamp, and double or through ligation was performed to secure the site. The abdominal cavity was then cleaned to ensure there was no bleeding at the ligation site, and the incision was closed. Post-operatively, the animals were housed individually. Generally, they were able to feed themselves after recovering from anesthesia and did not require special care.

We hope this detailed description addresses your queries and provides a clear understanding of the experimental conditions and procedures used in our study. These methodological details are crucial for ensuring the accuracy and reproducibility of our research findings.

(2) The lack of changes in MCH needs explanation? During stress erythropoiesis some limit in iron availability should cause MCH decrease particularly if the authors claim that macrophages for rapid iron recycling are decreased. Fig 1A is dispensable. Fig 1G NN control 14 days does not make sense since it is higher than 7 days of HH.

Thank you for your inquiry regarding the lack of changes in Mean Corpuscular Hemoglobin (MCH) in our study, particularly in the context of stress erythropoiesis and decreased macrophage-mediated iron recycling. We appreciate the opportunity to provide further clarification on this aspect.

(1) Explanation for stable MCH levels:

Our research identified a decrease in erythrophagocytosis and iron recycling in the spleen following HH exposure. Despite this, the MCH levels remained stable. This observation can be explained by considering the compensatory roles of other organs, particularly the liver and duodenum, in maintaining iron homeostasis.Specifically, our investigations revealed an enhanced capacity of the liver to engulf RBCs and process iron under HH conditions. This increased hepatic erythrophagocytosis likely compensates for the reduced splenic activity, thereby stabilizing MCH levels.

(2) Role of hepcidin and DMT1 expression:

Additionally, hypoxia is known to influence iron metabolism through the downregulation of Hepcidin and upregulation of Divalent Metal Transporter 1 (DMT1) expression. These alterations lead to enhanced intestinal iron absorption and increased blood iron levels, further contributing to the maintenance of MCH levels despite reduced splenic iron recycling.

(3) Revised Figure 1 and data presentation

To address the confusion regarding the data presented in Figure 1G, we have made revisions in our manuscript. The original Figure 1G, which did not align with the expected trends, has been removed. In its place, we have included a statistical chart of Figure 1F in the new version of Figure 1G. This revision will provide a clearer and more accurate representation of our findings.

(4) Manuscript updates and future research:

We update our manuscript to incorporate these explanations, ensuring that the rationale behind the stable MCH levels is clearly articulated. This includes a discussion on the role of the liver and duodenum in iron metabolism under hypoxic conditions.Future research could explore in greater detail the mechanisms by which different organs contribute to iron homeostasis under stress conditions like HH, particularly focusing on the dynamic interplay between hepatic and splenic functions.

We thank you for your insightful question, which has prompted a thorough re-examination of our findings and interpretations. We believe that these clarifications will enhance the overall understanding of our study and its implications in the context of iron metabolism and erythropoiesis under hypoxic conditions.

(3) Fig 2 the difference between sham and splenectomy is really marginal and not convincing. Is there also a difference at 7 days? Why does the spleen size decrease between 7 and 14 days?

Thank you for your observations regarding the marginal differences observed between sham and splenectomy groups in Figure 2, as well as your inquiries about spleen size dynamics over time. We appreciate this opportunity to clarify these aspects of our study.

(1) Splenectomy vs. Sham group differences:

In our experiments, the difference between the sham and splenectomy groups under HH conditions, though subtle, was consistent with our hypothesis regarding the spleen's role in erythrophagocytosis and stress erythropoiesis. Under NN conditions, no significant difference was observed between these groups, which aligns with the expectation that the spleen's contribution is more pronounced under hypoxic stress.

(2) Spleen size dynamics and peak stress erythropoiesis:

The observed splenic enlargement prior to 7 days can be attributed to a combination of factors, including the retention of RBCs and extramedullary hematopoiesis, which is known to be a response to hypoxic stress.Prior research has elucidated that splenic stress-induced erythropoiesis, triggered by hypoxic conditions, typically attains its zenith within a timeframe of 3 to 7 days. This observation aligns with our Toluidine Blue (TO) staining results, which indicated that the apex of this response occurs at the 7-day mark (as depicted in Figure 1, F-G). Here, the culmination of this peak is characteristically succeeded by a diminution in extramedullary hematopoiesis, a phenomenon that could elucidate the observed contraction in spleen size, particularly in the interval between 7 and 14 days.This pattern of splenic response under prolonged hypoxic stress is corroborated by studies such as those conducted by Wang et al. (2021), Harada et al. (2015), and Cenariu et al. (2021). These references collectively underscore that the spleen undergoes significant dynamism in reaction to sustained hypoxia. This dynamism is initially manifested as an enlargement of the spleen, attributable to escalated erythropoiesis and erythrophagocytosis. Subsequently, as these processes approach normalization, a regression in spleen size ensues.

We’ve revised our manuscript to include a more detailed explanation of these splenic dynamics under HH conditions, referencing the relevant literature to provide a comprehensive context for our findings. We will also consider performing additional analysis or providing further data on spleen size changes at 7 days to support our observations and ensure a thorough understanding of the splenic response to hypoxic stress over time.

(4) Fig 3 B the clusters should be explained in detail. If the decrease in macrophages in Fig 3K/L is responsible for the effect, why does splenectomy not have a much stronger effect? How do the authors know which cells died in the calcein stained population in Fig 3D?

Thank you for your insightful questions regarding the details of our data presentation in Figure 3, particularly about the identification of cell clusters and the implications of macrophage reduction. We appreciate the opportunity to address these aspects and clarify our findings.

(1) Explanation of cell clusters in Figure 3B:

In the revised manuscript, we have included detailed notes for each cell population represented in Figure 3B (Figure 3D in revised manuscript). These notes provide a clearer understanding of the cell types present in each cluster, enhancing the interpretability of our single-cell sequencing data.This detailed annotation will help readers to better understand the composition of the splenic cell populations under study and how they are affected by hypoxic conditions.

(2) Impact of splenectomy vs. macrophage reduction:

The interplay between the reduction in macrophage populations, as evidenced by our single-cell sequencing data, and the ramifications of splenectomy presents a multifaceted scenario. Notably, the observed decline in macrophage numbers following HH exposure does not straightforwardly equate to a comparable alteration in overall splenic function, as might be anticipated with splenectomy.In the context of splenectomy under HH conditions, a significant escalation in the RBCs count was observed, surpassing that in non-splenectomized mice exposed to HH. This finding underscores the spleen's critical role in modulating RBCs dynamics under HH. It also indirectly suggests that the diminished phagocytic capacity of the spleen following HH exposure contributes to an augmented RBCs count, albeit to a lesser extent than in the splenectomy group. This difference is attributed to the fact that, while the number of RPMs in the spleen post-HH is reduced, they are still present, unlike in the case of splenectomy, where they are entirely absent.Splenectomy entails the complete removal of the spleen, thus eliminating a broad spectrum of functions beyond erythrophagocytosis and iron recycling mediated by macrophages. The nuanced changes observed in our study may be reflective of the spleen's diverse functionalities and the organism's adaptive compensatory mechanisms in response to the loss of this organ.

(3) Calcein stained population in Figure 3D:

Regarding the identification of cell death in the calcein-stained population in Figure 3D (Figure 3A in revised manuscript), we acknowledge that the specific cell types undergoing death could not be distinctly determined from this analysis alone.The calcein staining method allows for the visualization of live (calcein-positive) and dead (calcein-negative) cells, but it does not provide specific information about the cell types. The decrease in macrophage population was inferred from the single-cell sequencing data, which offered a more precise identification of cell types.

(4) Revised manuscript and data presentation:

Considering your feedback, we have revised our manuscript to provide a more comprehensive explanation of the data presented in Figure 3, including the nature of the cell clusters and the interpretation of the calcein staining results.We have also updated the manuscript to reflect the removal of Figure 3K/L results and to provide a more focused discussion on the relevant findings.

We are grateful for your detailed review, which has helped us to refine our data presentation and interpretation. These clarifications and revisions will enhance the clarity and scientific rigor of our manuscript, ensuring that our conclusions are well-supported and accurately conveyed.

(5) Is the reduced phagocytic capacity in Fig 4B significant? Erythrophagocytosis is compromised due to the considerable spontaneous loss of labelled erythrocytes; could other assays help? (potentially by a modified Chromium release assay?). Is it necessary to stimulated phagocytosis to see a significant effect?

Thank you for your inquiry regarding the significance of the reduced phagocytic capacity observed in Figure 4B, and the potential for employing alternative assays to elucidate erythrophagocytosis dynamics under HH conditions.

(1) Significance of reduced phagocytic capacity:

The observed reduction in the amplitude of fluorescently labeled RBCs in both the blood and spleen under HH conditions suggests a decrease in erythrophagocytosis. This is indicative of a diminished phagocytic capacity, particularly when contrasted with NN conditions.

(2) Investigation of erythrophagocytosis dynamics:

To delve deeper into erythrophagocytosis under HH, we employed Tuftsin to enhance this process. Following the injection of PKH67-labeled RBCs and subsequent HH exposure, we noted a significant decrease in PKH67 fluorescence in the spleen, particularly marked after the administration of Tuftsin. This finding implies that stimulated erythrophagocytosis can influence RBCs lifespan.

(3) Erythrophagocytosis under normal and hypoxic conditions:

Under normal conditions, the reduction in phagocytic activity is less apparent without stimulation. However, under HH conditions, our findings demonstrate a clear weakening of the phagocytic effect. While we established that promoting phagocytosis under NN conditions affects RBC lifespan, the impact of enhanced phagocytosis under HH on RBCs numbers was not explicitly investigated.

(4) Potential for alternative assays:

Considering the considerable spontaneous loss of labeled erythrocytes, alternative assays such as a modified Chromium release assay could provide further insights. Such assays might offer a more nuanced understanding of erythrophagocytosis efficiency and the stability of labeled RBCs under different conditions.

(5) Future research directions:

The implications of these results suggest that future studies should focus on comparing the effects of stimulated phagocytosis under both NN and HH conditions. This would offer a clearer picture of the impact of hypoxia on the phagocytic capacity of macrophages and the subsequent effects on RBC turnover.

In summary, our findings indicate a diminished erythrophagocytic capacity, with enhanced phagocytosis affecting RBCs lifespan. Further investigation, potentially using alternative assays, would be beneficial to comprehensively understand the dynamics of erythrophagocytosis in different physiological states.

(6) Can the observed ferroptosis be influenced by bi- and not trivalent iron chelators?

Thank you for your question regarding the potential influence of bi- and trivalent iron chelators on ferroptosis under hypoxic conditions. We appreciate the opportunity to discuss the implications of our findings in this context.

(1) Analysis of iron chelators on ferroptosis:

In our study, we did not specifically analyze the effects of bi- and trivalent iron chelators on ferroptosis under hypoxia. However, our observations with Deferoxamine (DFO), a well-known iron chelator, provide some insights into how iron chelation may influence ferroptosis in splenic macrophages under hypoxic conditions.

(2) Effect of DFO on oxidative stress markers:

Our findings showed that under 1% O2, there was an increase in Malondialdehyde (MDA) content, a marker of lipid peroxidation, and a decrease in Glutathione (GSH) content, indicative of oxidative stress. These changes are consistent with the induction of ferroptosis, which is characterized by increased lipid peroxidation and depletion of antioxidants. Treatment with Ferrostatin-1 (Fer-1) and DFO effectively reversed these alterations. This suggests that DFO, like Fer-1, can mitigate ferroptosis in splenic macrophages under hypoxia, primarily by impacting MDA and GSH levels.

**Author response image 17. sa3fig17:** 

(3) Potential role of iron chelators in ferroptosis:

The effectiveness of DFO in reducing markers of ferroptosis indicates that iron availability plays a crucial role in the ferroptotic process under hypoxic conditions. It is plausible that both bi- and trivalent iron chelators could influence ferroptosis, given their ability to modulate iron availability within cells. Since ferroptosis is an iron-dependent form of cell death, chelating iron, irrespective of its valence state, could potentially disrupt the process by limiting the iron necessary for the generation of reactive oxygen species and lipid peroxidation.

(4) Additional research and manuscript updates:

Our study highlights the need for further research to explore the differential effects of various iron chelators on ferroptosis, particularly under hypoxic conditions. Such studies could provide a more comprehensive understanding of the role of iron in ferroptosis and the potential therapeutic applications of iron chelators. We update our manuscript to include these findings and discuss the potential implications of iron chelation in the context of ferroptosis under hypoxic conditions. This will provide a broader perspective on our research and its significance in understanding the mechanisms of ferroptosis.